# SARS-CoV-2 infection establishes a stable and age-independent CD8+ T cell response against a dominant nucleocapsid epitope using restricted T cell receptors

The resolution of SARS-CoV-2 replication hinges on cell-mediated immunity, wherein CD8+ T cells play a vital role. Nonetheless, the characterization of the specificity and TCR composition of CD8+ T cells targeting non-spike protein of SARS-CoV-2 before and after infection remains incomplete. Here, we analyzed CD8+ T cells recognizing six epitopes from the SARS-CoV-2 nucleocapsid (N) protein and found that SARS-CoV-2 infection slightly increased the frequencies of N-recognizing CD8+ T cells but significantly enhanced activation-induced proliferation compared to that of the uninfected donors. The frequencies of N-specific CD8+ T cells and their proliferative response to stimulation did not decrease over one year. We identified the $N_{222-230}$ peptide (LLLDRLNQL, referred to as LLL thereafter) as a dominant epitope that elicited the greatest proliferative response from both convalescent and uninfected donors. Single-cell sequencing of T cell receptors (TCR) from LLL-specific CD8+ T cells revealed highly restricted Vα gene usage (TRAV12-2) with limited CDR3α motifs, supported by structural characterization of the TCR–LLL–HLA-A2 complex. Lastly, transcriptome analysis of LLL-specific CD8+ T cells from donors who had expansion (expanders) or no expansion (non-expanders) after in vitro stimulation identified increased chromatin modification and innate immune functions of CD8+ T cells in non-expanders. These results suggests that SARS-CoV-2 infection induces LLL-specific CD8+ T cell responses with a restricted TCR repertoire.

CD8+ T cells play a vital role in combatting SARS-CoV-2 and forming long-term memory responses to this coronavirus[1–3]. Unlike the viral epitopes recognized by antibodies, which are sensitive to mutations causing viral escape by new variants, CD8+ T cells recognize epitopes from both mutable and highly conserved viral proteins, offering longer immune protection[4,5]. Due to the complex nature of antigen recognition by T cell receptors (TCR), which involves the presentation of many epitopes by highly polymorphic human leukocyte antigen (HLA) molecules, TCR repertoires for defined SARS-CoV-2 epitopes have not been as fully characterized as antibody repertoires.

Activation of CD8+ T cells is observed in the blood of COVID-19 patients[6,7], and low CD8+ T cell counts are associated with severity of COVID-19 symptoms and poor outcomes[8–10]. Analysis of the targets recognized by CD8+ T cells after in vitro stimulation with pooled peptides of SARS-CoV-2 proteins has shown recognition of both highly conserved structural proteins, such as nucleocapsid (N) and membrane proteins, as well as the highly mutable spike (S) protein of SARS-

✉ e-mail: wengn@mail.nih.gov

CoV-2[11–16]. Furthermore, CD8+ T cells recognizing SARS-CoV-2 are not only found in COVID-19 patients and vaccinated donors but also in uninfected donors[7,12,17,18]. Phenotypically, both naïve and memory subsets exist in SARS-CoV-2-recognizing CD8+ T cells of COVID-19 patients, vaccinated donors, and uninfected individuals. Acute SARS-CoV-2 infection generates memory T cells[19–21], but the exact functional changes in these memory T cells remain to be determined. The presence of differentiated CD8+ T cells recognizing epitopes from the N protein in donors without any known prior SARS-CoV-2 infection suggests that these CD8+ T cells are likely cross-reactive to other common coronaviruses.

Previous studies of CD8+ T cell responses to SARS-CoV-2 have focused mainly on epitopes derived from the S protein. For example, S$_{269-277}$ (YLQPRTFLL, referred to as YLQ) is a dominant yet variable spike epitope that elicits a polyfunctional CD8+ T cell response in COVID-19 recovered patients[13,15,22]. Sequence analysis of YLQ-specific TCR repertoire revealed public TCRs with highly biased usage of the TRAV12-1 and TRAV12-2 gene segments[23]. Crystal structures of TCR–YLQ–HLA-A2 complexes provided insights into the selection of particular TRAV and TRBV genes and the effects of viral variants on TCR recognition[23–26]. Less is known about the TCR repertoires elicited by nucleocapsid epitopes[14,27]. N$_{222-230}$ (LLLDRNQL, referred to as LLL) is presented by HLA-A2 and has a broad CD8+ T cell recognition by peptide stimulation and tetramer staining[11]. Of note, LLL is one of six SARS-CoV-2 T cell epitopes included in a recent peptide-based vaccine against COVID-19 (CoVac-1)[28,29]. This vaccine induced T cell responses in a Phase I/II clinical trial that were unaffected by current SARS-CoV-2 variants of concern. The LLL peptide is also a component of a T cell-directed mRNA vaccine (BNT162b4) that protected hamsters against severe disease[30].

Aging is associated with changes in CD8+ T cell homeostasis and functions[31,32], including a reduction in circulating naïve CD8+ T cells and an increase in differentiated memory CD8+ T cells due to thymic atrophy and lifelong stimulation by environmental and intrinsic insults[33–36]. This leads to reduced TCR repertoire diversity in older adults[36,37] and decreased immune response to various infections and vaccines[38–40]. Despite the high mortality rate of COVID-19[41,42], older adults tend to have fairly robust antibody responses to the mRNA-based COVID-19 vaccines[43,44]. Analysis of CD8+ T cells in patients with acute COVID-19 has shown that reduced naïve T cells and reduced antigen-specific T cell responses are observed in older patients with severe COVID-19[42,45]. The mechanism behind reduced T cell function with age has recently been analyzed using high-dimensional flow cytometry and multi-omics data[34], but it is still unclear what changes in CD8+ T cells determine their activation-induced proliferation and function.

In this study, we analyzed the frequency, differentiation status, and in vitro expansion of circulating CD8+ T cells recognizing six epitopes from the SARS-CoV-2 N protein in uninfected and convalescent COVID-19 donors. We found that the frequencies of CD8+ T cells recognizing the LLL epitope were significantly higher in recovered patients than in uninfected donors and remains stable over one year follow-up. In vitro antigenic challenge identified LLL-specific CD8+ T cells from convalescent donors had a significantly higher percentage of expanders and a more robust proliferative response than uninfected donors. Further scTCRseq analysis of LLL-specific CD8+ T cells showed highly restricted Vα gene usage (TRAV12-2) with limited CDR3α motifs, supported by the crystal structure of a TCR–LLL–HLA-A2 complex. Lastly, single-cell transcriptome analysis of LLL-specific CD8+ T cells from donors who had expansion or no expansion after in vitro stimulation identified increased chromatin modification and innate immune functions of CD8+ T cells from non-expanders in a TCR-independent manner, suggesting that these transcriptome changes may regulate activation-induced CD8+ T cell proliferation and expansion.

## Results

### Increased frequencies of SARS-CoV-2 nucleocapsid-specific CD8+ T cells post-infection

To understand CD8+ T cell immunity against the highly conserved SARS-CoV-2 nucleocapsid protein, we analyzed the frequencies of N-recognizing circulating CD8+ T cells from COVID-19 convalescent donors (mild clinical presentations of the disease and not hospitalized, $n = 75$, F = 56, M = 19, age range 18–89 years old) and uninfected controls ($n = 138$, F = 74, M = 64, age 17–92 years old) (Fig. 1a, Supplementary Data 1). Convalescent donors had a positive PCR test and detectable levels of anti-N IgG antibodies (Fig. 1b) or detectable anti-S IgG antibodies. The uninfected controls had undetectable levels of blood anti-N IgG antibodies (Fig. 1b). Utilizing multi-color flow cytometry, we measured three differentiation markers (CD127, CD28, and CD27) within the general CD8+ T cell population in the peripheral blood of convalescent donors and uninfected controls. Convalescent donors had significantly increased percentages of CD8+ T cells expressing CD127+ (IL7R) and CD28+/CD27+ and reduced CD28−/CD27+ subsets compared to the uninfected controls (Fig. 1c, Supplementary Fig. 1a). These findings suggest that SARS-CoV-2 infection alters the composition and status of CD8+ T cells through an enrichment of cells that are not fully differentiated.

We analyzed CD8+ T cells recognizing six previously reported epitopes (presented by HLA-A2) of the highly conserved N protein of SARS-CoV[46,47] in convalescent ($n = 34–35$) and uninfected controls ($n = 21–73$) who are HLA-A2 positive. To measure the frequency of circulating N-specific CD8+ T cells, we created six MHC class I tetramers (HLA-A2) bearing these six epitopes (Fig. 1d). In agreement with previous reports[11,16,22,48], we found that both convalescent and uninfected donors had less than 1% of CD8+ T cells positive for each tetramer; however, compared to their uninfected counterparts, COVID-19 convalescent patients had significantly higher frequencies of CD8+ T cells specific for the epitope LLL and for the sum of all six epitopes (Fig. 1e). Specifically, central memory (T$_{CM}$) CD8+ T cells specific for LLL was increased in convalescent patients (Fig. 1f). Although there is no substantial sequence similarity between these epitopes of SARS-CoV-2 and other common coronaviruses (Supplementary Table 1)[49], these memory phenotype N-epitope recognizing CD8+ T cells may derive from cross-reactive TCRs with other antigens. Collectively, we observed that SARS-CoV-2 infection is associated with significantly higher frequencies of both LLL-specific and sum of six N epitope-specific CD8+ T cells compared to uninfected controls.

### Stable frequency of nucleocapsid-specific CD8+ T cells over time

To examine changes in the frequency and function of CD8+ T cells over time, we collected samples from convalescent donors across three visits over a year (Fig. 2a). As previously reported[50], we observed a significant decline of anti-nucleocapsid protein antibody titers in the blood, with a decay rate of −0.012 AU/mL per day ($p = 0.001$) (Fig. 2b). In contrast, the frequency of CD8+ T cells recognizing the six epitopes of nucleocapsid protein did not reduce over the course of a year (Fig. 2c–e, Supplementary Fig. 2a). We further analyzed changes in the subsets of LLL-specific CD8+ T cells and found that T$_{CM}$ percentage significantly increased over time, but the other four subsets (T$_N$, T$_{SCM}$, T$_{EM}$, and T$_{EMRA}$) were not significantly changed (Supplementary Fig. 2b). These findings revealed that, in contrast to the decline of anti-nucleocapsid IgG titer over the course of a year, the overall frequencies and proportions of different subsets of nucleocapsid-specific CD8+ T cells remain stable.

### SARS-CoV-2 infection is associated with enhanced activation and proliferation of N-specific CD8+ T cells in vitro

To study how N-recognizing CD8+ T cells respond to antigenic challenge, we performed overnight stimulation with a pool of peptides from S and N protein (ALN, LQL, LLL, GMS, ILL, and LAL) and checked

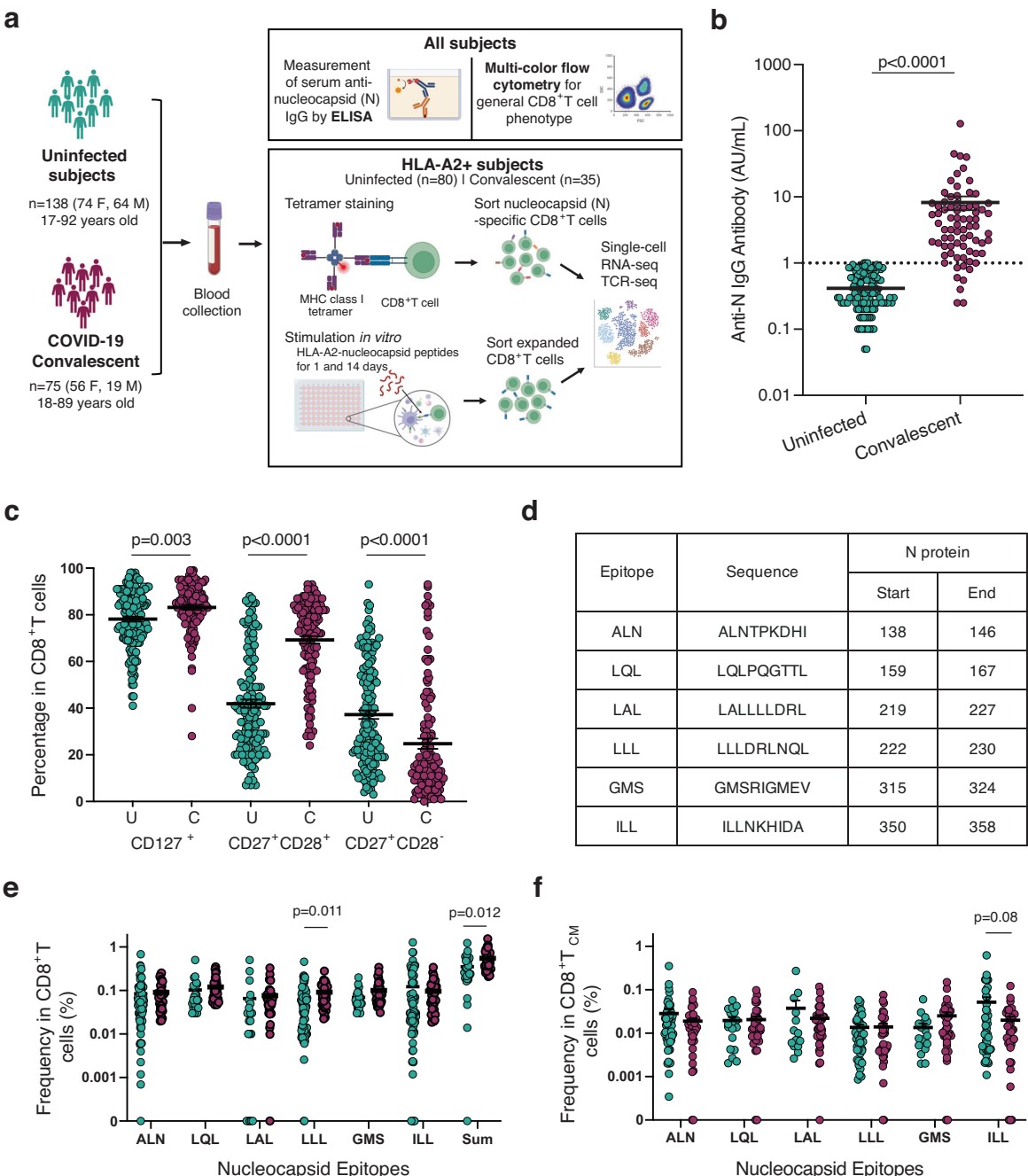

**Fig. 1 | Experimental scheme and frequencies of CD8⁺ T cell recognizing six epitopes of nucleocapsid protein of SARS-CoV-2. a** Overview of experimental design. Image created using BioRender.com. **b** Anti-nucleocapsid IgG titers in uninfected and convalescent donors. Anti-nucleocapsid IgG were measured using ELISA. Samples with concentrations below 1.0 AU/mL were undetectable for IgG antibodies (convalescent = 75 and uninfected = 138). **c** Significant differences in CD127, CD28, and CD27 expression in CD8⁺ T cells between uninfected and convalescent donors (C = Convalescent, $n = 118$, U = Uninfected, $n = 56$). PBMCs were isolated from blood and were stained with a panel of 15 antibodies. **d** Information on six SARS-CoV-2 nucleocapsid epitopes. **e** Frequencies of CD8⁺ T cells recognizing six nucleocapsid epitopes in uninfected and convalescent donors (Convalescent = 35 and Uninfected = 80). Each epitope specific tetramer was used to measure the frequency of epitope recognizing CD8⁺ T cells in PBMCs by flow cytometry. The sum refers to the summation of the frequencies of CD8⁺ T cells for all six nucleocapsid epitopes. **f** Frequencies of CD8⁺ T cell central memory (T_CM defined by CD62⁺CD45RA⁻) subset that recognizes six nucleocapsid epitopes (Convalescent $n = 35$, Uninfected $n = 80$). Two-tailed $T$ test adjusted for age and sex were was carried out for all comparisons between convalescent and uninfected donor. $p$ value, mean and SEM are shown.

for any upregulation of activation-induced markers (Fig. 3a). We observed a mild increase in CD69⁺ CD8⁺ T cells in convalescent donors compared to uninfected controls (Supplementary Fig. 3a). Activation marker expression, however, does not necessarily predict if downstream expansion of CD8⁺ T cells will occur, so to measure this, we performed a long-term in vitro stimulation to observe the expansion of

CD8⁺ T cells specific to these six nucleocapsid epitopes (Fig. 3a). We found that convalescent patients were able to expand in response to all six nucleocapsid epitopes, while uninfected donors could only respond to four epitopes (LLL, ALN, LQL, and ILL) (Fig. 3b, Supplementary Fig. 3b, and Supplementary Table 2). Among the six epitopes, we observed that LLL induced CD8⁺ T cell expansion in 94% of

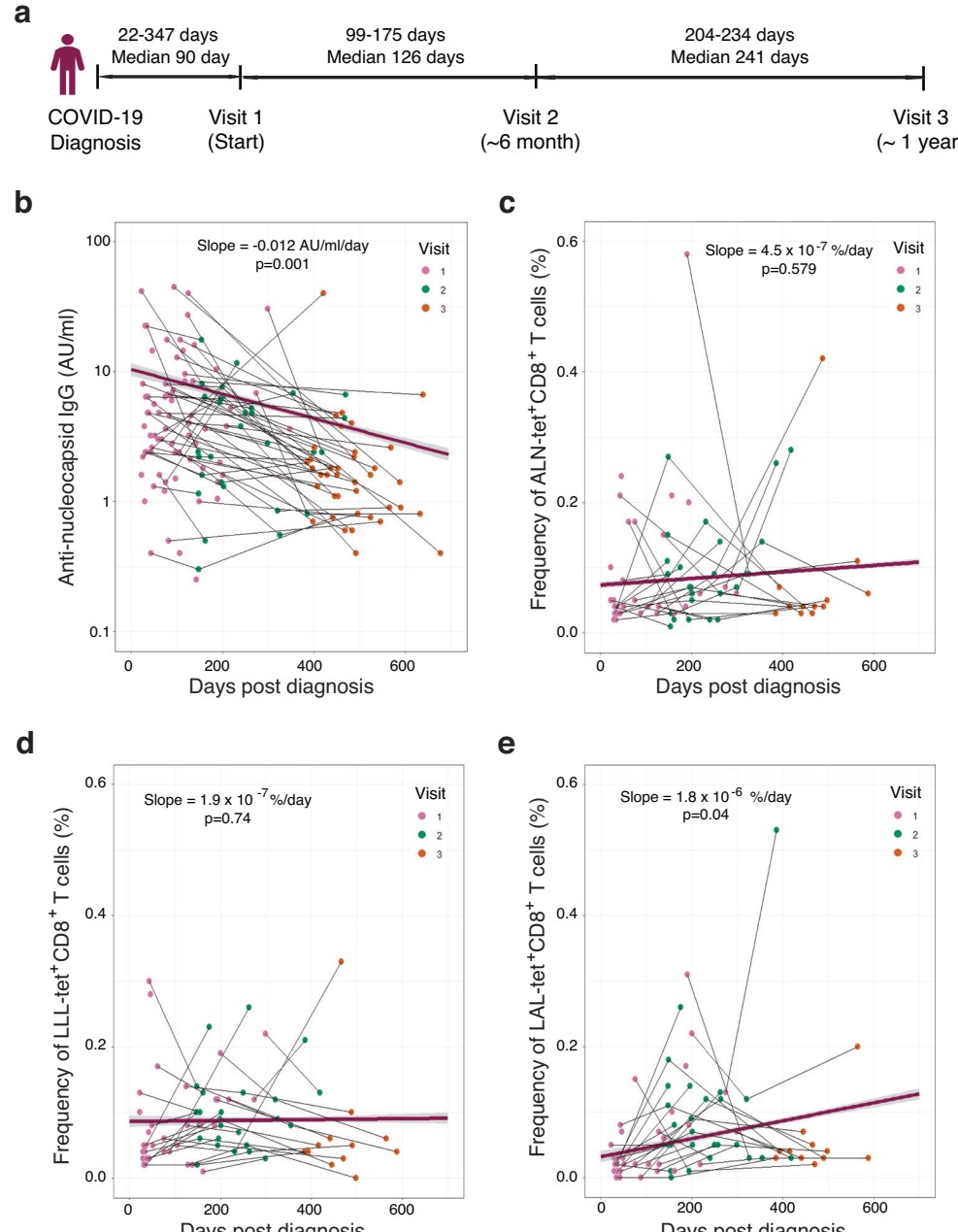

**Fig. 2 | Longitudinal analysis of LLL-specific CD8⁺ T cells in the blood of convalescent subjects. a** Overview of the approximate timepoints for multiple visits of convalescent subjects. **b** Anti-nucleocapsid IgG titers at multiple time points (pink = visit 1, green = visit 2, orange = visit 3 used for this figure). Subjects with multiple visits are connected with black trendlines. The overall trend line (purple) across time points for each subject were generated by the mixed effect linear regression model (adjusted for age and sex) for the convalescent cohort with error bounds indicating a 95% confidence interval (light gray) $n = 145$. **c** Frequency of ALN-tetramer positive CD8⁺ T cells at multiple time points (red = visit 1, green = visit 2, blue = visit 3). Multiple visits for a single subject are connected by thin black lines. The overall purple trend line across time points for each subject were generated by the mix effect linear regression model (adjusted for age and sex) for the convalescent cohort with error bounds indicating a 95% confidence interval (light gray), $n = 78$. **d** Frequency of LLL-tetramer positive CD8⁺ T cells at multiple time points. **e** Frequency of LAL-tetramer positive CD8⁺ T cells at multiple time points.

convalescent patients, compared to 50% of uninfected donors. To quantify the degree of expansion, we set the classifications as mild ($1 < x < 3$ cell divisions) or robust ($\geq 3$ cell divisions). We found that the average LLL-induced CD8⁺ T cell expansion in convalescent donors was four divisions higher (cell count 16-fold higher) compared to uninfected donors (Fig. 3c). These findings suggest that convalescent donors have a greater ability to expand CD8⁺ T cells in response to nucleocapsid epitopes, particularly to the dominant LLL epitope, compared to uninfected donors.

To determine whether the initial number of LLL-recognizing CD8⁺ T cells and their differentiation status influence activation-induced LLL-specific cell expansion, we compared the number of seeded naive and memory epitope-recognizing CD8⁺ T cells and the magnitude of expansion after in vitro stimulation. We found that the magnitude of expansion of LLL⁺ CD8⁺ T cells was positively correlated with the number of initially seeded $T_{EM}$ LLL⁺ CD8⁺ T cells ($p = 0.009$) (Fig. 3d). The number of seeded LLL-specific CD8⁺ T cells of $T_N$, $T_{SCM}$, $T_{CM}$, and $T_{EMRA}$ differentiation status had no effect on the magnitude of LLL⁺

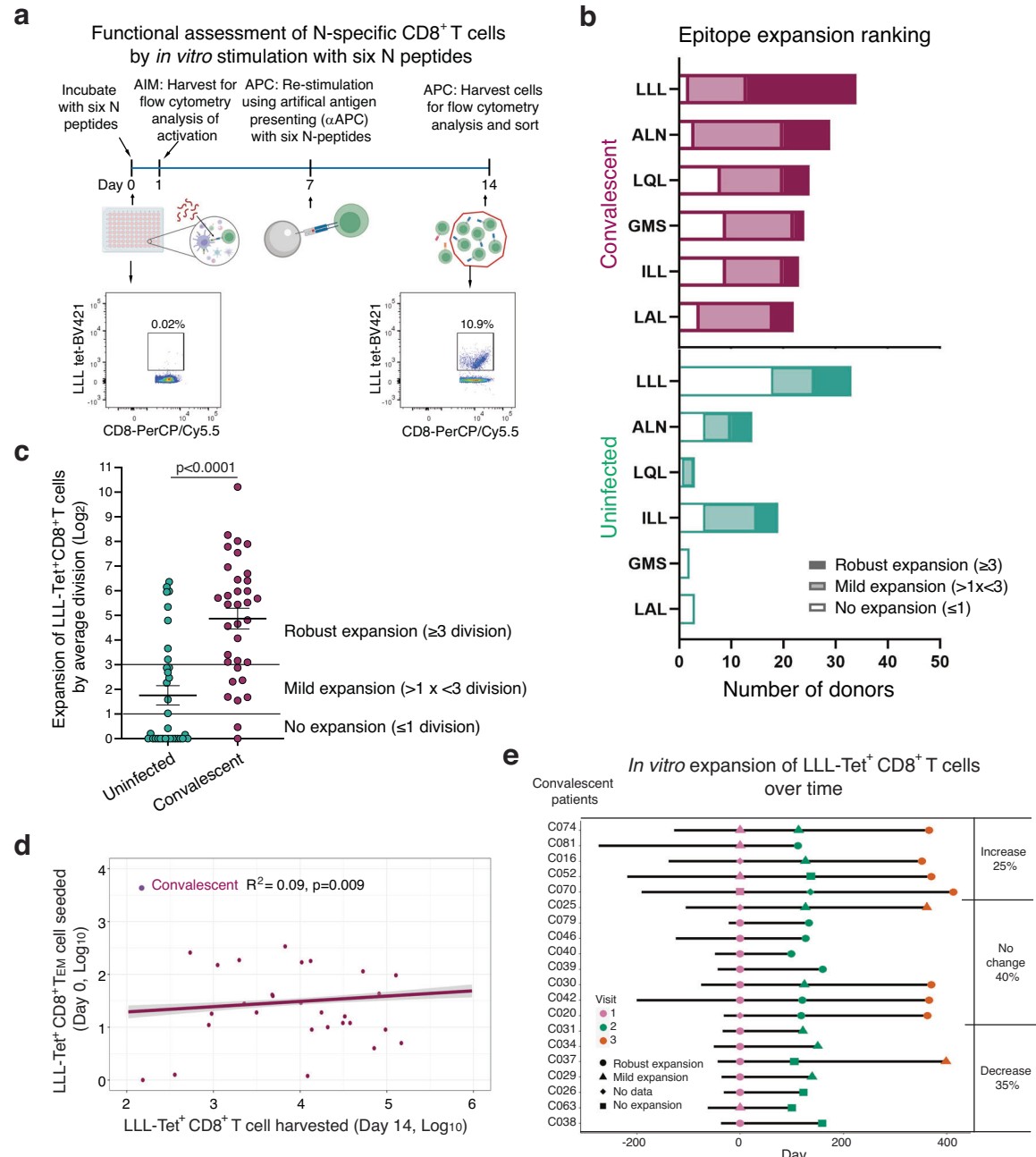

**Fig. 3 | Functional assessment of the ability for in vitro expansion of SARS-CoV-2 nucleocapsid-specific CD8⁺ T cells. a** Experimental overview of functional assessment of in vitro antigen-induced activation and expansion of nucleocapsid-specific CD8⁺ T cells. AIM = activation induced marker measurement and APC = artificial antigen presentation complex assay. Image created using BioRender.com. **b** Ranking based on the percentage and degree of epitope-specific CD8⁺ T cell expansion post fourteen-day stimulation with nucleocapsid epitopes. Expansion is categorized based on the fold change of antigen-specific cells after fourteen days (none≤1, mild 1 <x< 3, robust ≥3 divisions). **c** Fold change of LLL-specific CD8⁺ T cells after stimulation. Each point represents an average value of each convalescent donor stimulated with LLL peptide-HLA-A2 (Convalescent $n = 34$, Uninfected $n = 43$), Two-tailed T-test adjusted for age and sex was carried out for all CD8⁺ T cell expansion (Supplementary Fig. 3c). To further determine whether activation-induced expansion changes over time, we compared the magnitude of in vitro expansion of LLL⁺ CD8⁺ T cells across multiple visits of convalescent donors. We found that expansion of LLL⁺ CD8⁺ T cells in the majority of donors (65%) was unchanged or comparisons between convalescent and uninfected donor. p value, mean and SEM are shown. **d** Correlation between seeding LLL-specific T_EM cell counts on day 0 and harvested LLL-specific CD8⁺ T cell counts after stimulation. Values were log₁₀ transformed and the purple trend line across all data for each subject were generated by the mixed effect linear regression model (adjusted for age and sex) with error bounds indicating a 95% confidence interval (light gray) showing the correlation of cell numbers between day 0 and day 14 ($R^2 = 0.09$, $p = 0.009$). **e** Stability of in vitro expansion of LLL-specific CD8⁺ T cells over time. Expansion of LLL-specific CD8⁺ T cells of convalescent donors were placed into three categories (increased expansion, decreased expansion, no major change in expansion) based on changes in expansion across different visits among three designated degrees of expansion (no, mild and robust expansion).

increased (Fig. 3e). Taken together, these findings identified a dominant nucleocapsid epitope (LLL) for CD8⁺ T cells in both convalescent and uninfected donors and demonstrated that SARS-CoV-2 infection primes LLL⁺ CD8⁺ T cells to have improved long-lasting expansion capabilities in response to subsequent antigenic challenge.

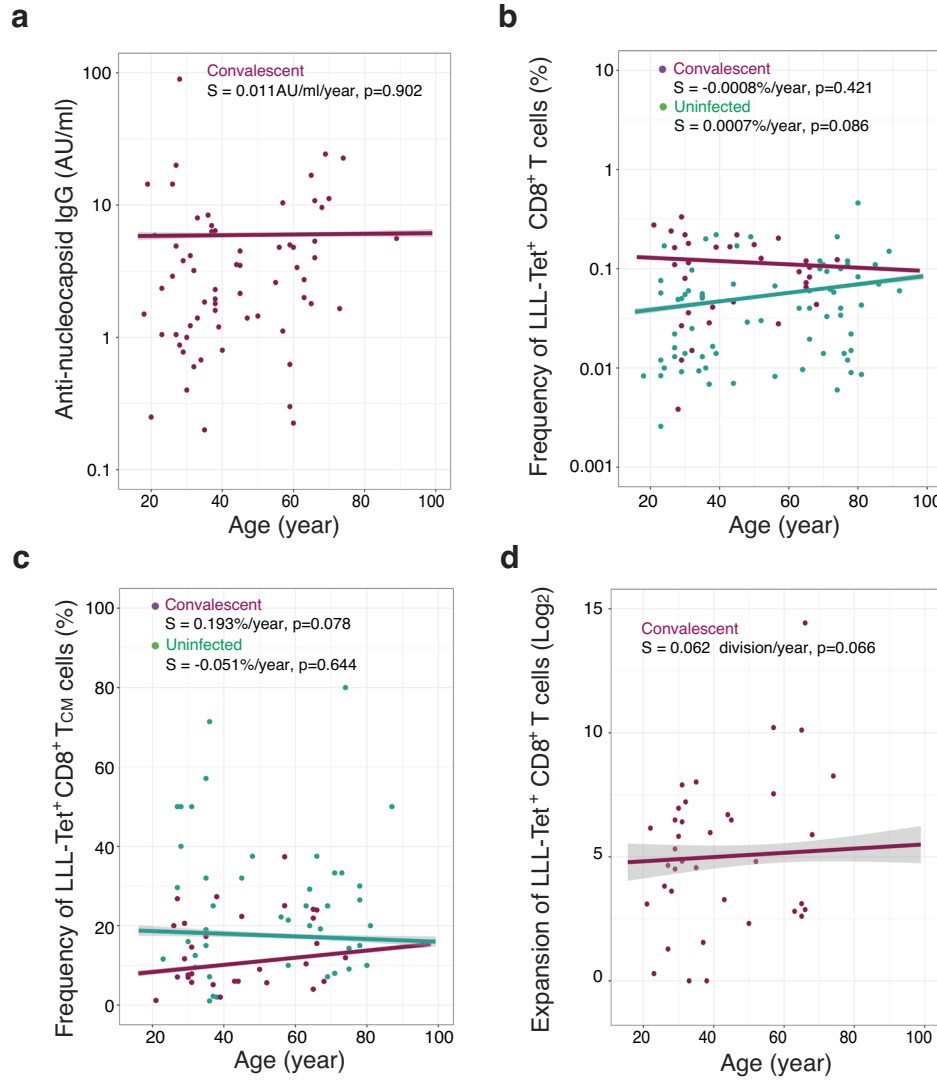

**Fig. 4 | Age-associated changes in titer of anti-nucleocapsid antibody and LLL-specific CD8+ T cells in uninfected and convalescent donors. a** Average anti-nucleocapsid IgG antibody titer was no obvious change with age in convalescent donors. For donors with multiple visits, the concentrations were averaged and depicted as a single point. The correlation between anti- nucleocapsid IgG antibody and age are shown by the pink trendline using the mixed effect linear regression model adjusted with sex. Error bounds indicating a 95% confidence interval (light gray) (*n* = 74). S = slope used in this figure. **b** Correlation between frequency of LLL-specific CD8+ T cells and donor's age. For donors with multiple visits, the frequencies were averaged and depicted as a single point. Pink line represents convalescent donors and green line represent uninfected donors, and they were

generated using the mixed effect linear regression model adjusted with sex. **c** Correlation between frequency of LLL-Tet+ CD8+ T$_{CM}$ cells and donor's age. For donors with multiple visits, the frequencies were averaged and depicted as a single point. Purple line represents convalescent donors and green line represent uninfected donors, and they were generated using the mixed effect linear regression model adjusted with sex. **d** Correlation between degree of LLL Tet+ CD8+ T cell expansion and donor's age. For donors with multiple visits, the expansions were averaged and depicted as a single point. Purple line represents convalescent donors which was generated using the mixed effect linear regression model adjusted with sex.

## Stable frequency and expansion of LLL-recognizing CD8+ T cells with age

Since COVID-19 disproportionately affects the elderly population, we sought to determine whether age alters the frequency and expansion of CD8+ T cells against SARS-CoV-2 N epitopes. We found that age does not affect the levels of anti-N IgG titer in convalescent donors (Fig. 4a), nor does it affect the frequencies of CD8+ T cells specific for six N epitopes in either convalescent or uninfected controls (Fig. 4b and Supplementary Fig. 4a). We further analyzed the different subsets of LLL+ CD8+ T cells and did not observe significant change in the frequency of LLL+ CD8+ T cell subsets (T$_N$, T$_{CM}$, T$_{SCM}$, T$_{EM}$, and T$_{EMRA}$) in convalescent donors with age (Fig. 4c and Supplementary Fig. 4b). We also did not find that the age of convalescent donors impacts the activation-induced expansion of CD8+ T cells against LLL or any of the

other five epitopes of the N protein (Fig. 4d and Supplementary Fig. 4c). Overall, age does not have a significant impact on anti-N IgG titer or LLL+ CD8+ T cell frequency and expansion; however, we acknowledge that the older convalescent donors in our study presented only mild symptoms of COVID-19 and thus may not reflect the immune status of the general elderly population.

## Highly restricted Vα gene usage by LLL-specific CD8+ TCRs

To investigate the α and β chain sequences of LLL-recognizing TCRs, we isolated LLL tetramer+ CD8+ T cells from 15 donors (6 convalescent and 7 uninfected donors) before stimulation and from 14 donors (13 convalescent and 1 uninfected donors) with LLL+ CD8+ T cells which expanded after in vitro stimulation. After sorting LLL+ CD8+ T cells using flow cytometry, we determined the TCR sequences of these

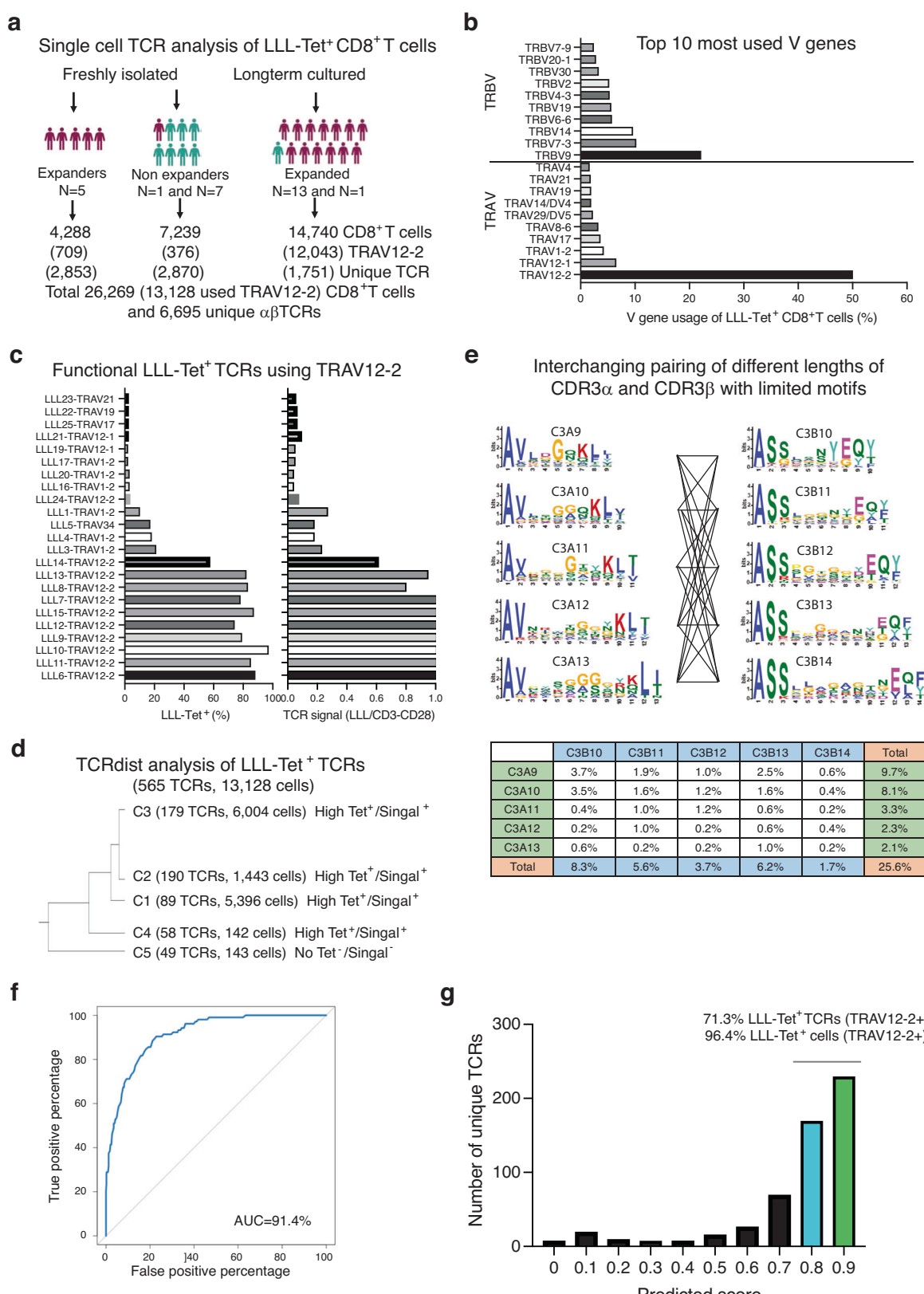

isolated cells through scTCR-seq of a total of 26,269 LLL⁺ CD8⁺ T cells (24,795 are primary α chain and 1474 additional TCRs containing a second unique functional α chain) consisting of 6,695 unique αβTCR sequences (Fig. 5a). Initial V gene usage analysis revealed a dominant Vα gene (TRAV12-2 = 50%) with relatively diverse Vβ genes (the most abundant: TRVB9 = 21%) in LLL⁺ CD8⁺ T cells (Fig. 5b). To confirm

binding specificity, we selected 23 LLL-recognizing TCRs and expressed them in a Jurkat T cell line (Fig. 5c, Supplementary Fig. 5a). We tested each TCR-expressing cell's ability to bind to the LLL-tetramer and checked for expression of Nur77, an early activation-induced gene expressed post TCR signaling, after LLL-HLA-A2 stimulation in vitro. We found that nine of the ten TCRs from the TRAV12-2 family (91%)

**Fig. 5 | Characteristics of TCRs and their predictability of binding to a dominant nucleocapsid LLL epitope. a** Summary of LLL tetramer⁺ CD8⁺ T cells isolated from uninfected, COVID-19 convalescent donors, and in vitro expanded donors by cell sorting and scTCR-seq. **b** Top ten most used V genes of LLL⁺ CD8⁺ T cells. Histogram depicting the top ten most used V-genes (Vα and Vβ) in all LLL⁺ CD8⁺ T cells. **c** Validation of LLL-HLA-A2 binding and signaling of cloned selected LLL⁺TCRs. Tetramer binding percentage and LLL-HLA-A2 stimulation induced signaling presented by the ratio of GFP% induced by LLL-HLA-A2 over anti-CD3/CD28 stimulation post 4 h. **d** Classification of LLL-TCRs using TRAV12-2 by TCRDist. Dendrogram of clusters of LLL-TCR (TRAV-12-2) (N = 575 with percentage of each cluster) defined by TCRDist. Cluster 1,2,4 and 5 containing confirmed LLL-TCRs

(N = 524) while cluster 3 did not (N = 51). **e** CDR3α and CDR3β motifs and their pairings of cluster 2. Motifs of each CDR3 length were generated by MEME Suite 5.5. The percentages of each combination between a CDR3α and a CDR3β motif in total number of LLL-TCRs (TRAV12-2) are presented. **f** Machine learning (ML) algorithm for predicting LLL-HLA-A2 binding TCRs. ROC of the predictive ability on unexposed testing TCRs of the ML algorithm for LLL-HLA-A2 TCR binding is 92.2%. **g** Proportion of ML predicted LLL-HLA-A2 binding TCRs in unique LLL⁺ TCRs (TRAV12-2) and total LLL⁺ (TRAV12-2) CD8⁺ T cells. The percentages of ML predicted LLL⁺ TCRs with scores greater than 0.8 are shown in unique LLL⁺ TCRs (TRAV12-2) and in total LLL⁺ (TRAV12-2) CD8⁺ T cells.

displayed strong binding to the LLL-tetramer as well as strong GFP signaling, but TCRs made up of the other five Vα gene families (TRAV12-1, 17, 19, 21, and 34) did not show a high percentage of tetramer binding nor had substantial activation-induced Nur77 reporter expression (Fig. 5c). We further analyzed LLL-specific TCRs used TRAV12-2 gene by TCRDist classification[51] and identified four clusters containing experimentally proved LLL-binding TCRs (N = 516 representing 13,037 cells) and one cluster contained a no LLL binding TCR (N = 49 representing 91 cells) (Fig. 5d). We then analyzed the CDR3 motifs within each cluster of LLL-binding TCRs and found that CDR3α had a limited number of motifs compared to CDR3β. CDR3 motifs of different lengths can be interchangeably paired and the combination of five CDR3α motifs and five CDR3β motifs of Cluster 2 (C2) accounted for 23% of LLL⁺TCR-TRAV12-2 (Fig. 5e). Clusters 1 and 3 (C1 and C3) had similar motif combinations and accounted for 22% and 10% of LLL⁺TCR-TRAV12-2, respectively (Supplementary Fig. 5b). Our findings revealed highly restricted Vα gene usage by LLL-specific TCRs and highly interchangeable pairing of TCRα and TCRβ within the TCR cluster.

Since approximately 43% (10/23) of the tested LLL-TCRs from the sorted LLL tetramer⁺ CD8⁺ T cells bound to LLL-HLA-A2 and delivered signals post-activation, we sought to develop a method to identify TCRs that bind to the LLL-HLA-A2 complex by using a random forest (RF) algorithm to score the TCRs based on their CDR3α and CDR3β sequences. We selected the positive TCRs from TRAV12-2⁺ LLL tetramer⁺ CD8⁺ T cells that were clustered by TCRDist and were confirmed to bind to LLL-HLA-A2 in vitro. In parallel, we used TCRDist to cluster TCRs specific to other SARS-CoV-2 epitopes besides LLL and selected the TCRs with no LLL-HLA-A2 binding as negative TCRs. (Supplementary Data 2). The amino acid sequences of CDR3α and CDR3β of both positive and negative TCRs were then broken down into 3-mers and given five positional encoding (left end, left, center, right, and right end) were trained by a RF model as described[52] (Supplementary Fig. 5c). The RF algorithm showed good accuracy in predicting unseen data, with an AUC (area under the curve) of 92.2% (Fig. 5f). Further analysis identified kmers that had the most impact on RF (Supplementary Fig. 5d). When we applied this RF algorithm to score all unique TRAV12-2⁺ TCRs from LLL tetramer⁺ sorted CD8⁺ T cells, we found that 76.5% of the TCRs had a score greater than 0.8, and this fraction of TCRs accounted for 91.1% of all TRAV12-2⁺ TCRs expressing CD8⁺ T cells (Fig. 5g). These results show that LLL-HLA-A2 binding TCRs preferentially use the TRAV12-2 gene and consist of a limited number of CDR3α motifs that are interchangeably paired with CDR3β motifs. Lastly, our machine learning (ML) algorithm demonstrates accurate prediction of LLL-HLA-A2 binding TCRs.

### Affinity and overall structure of an LLL-specific TCR bound to LLL−HLA-A2

The above findings led us to investigate the structural basis for dominant usage of the TRAV12-2 gene segment by LLL-specific TCRs. TCR LLL8, which utilizes TRAV12-2 and TRAJ54 for the α chain and TRBV7-2 and TRBJ2-1 for the β chain, was selected for further characterization. We used surface plasmon resonance (SPR) to measure the

affinity of TCR LLL8 for HLA-A2 loaded with LLL peptide (Fig. 6a). TCR LLL8 bound LLL−HLA-A2 with a dissociation constant ($K_D$) of $19.2 \pm 1.2\,\mu M$. This affinity is within the range of TCRs specific for microbial antigens ($1–50\,\mu M$)[53], including TCRs specific for SARS-CoV-2 spike epitopes[25,54]. Kinetic parameters (on- and off-rates) for the binding of LLL8 to LLL−HLA-A2 were $k_{on} = 1.7 \times 10^4\,M^{-1}s^{-1}$ and $k_{off} = 0.34\,s^{-1}$, corresponding to a $K_D$ of $20.4\,\mu M$ (Fig. 6a), in close agreement with the $K_D$ from equilibrium analysis ($19.2\,\mu M$).

To understand how LLL-specific TCRs isolated from COVID-19 convalescent patients recognize the LLL epitope, we determined the structure of the LLL8−LLL−HLA-A2 complex to 3.18 Å resolution (Fig. 6b, Supplementary Table 3). The interface between TCR and pMHC was in unambiguous electron density for each of the four complex molecules in the asymmetric unit of the crystal (Supplementary Fig. 6a). The root-mean-square difference (r.m.s.d.) in α-carbon positions for the TCR VαVβ and MHC α1α2 modules, including the LLL peptide, ranged from 0.5 Å to 1.0 Å for the four LLL8−LLL−HLA-A2 complexes, indicating close similarity. Therefore, the following description of TCR–pMHC interactions applies to all molecules in the asymmetric unit of the crystal unless noted otherwise. TCR LLL8 docks over LLL−HLA-A2 in a canonical diagonal orientation, with Vα over the α2 helix of HLA-A2 and Vβ over the α1 helix. The crossing angle of TCR to pMHC[55] is 31° (Fig. 6c). The incident angle[56], which corresponds to the degree of tilt of TCR over pMHC, is 3°. As depicted by the footprint of TCR LLL8 on the pMHC surface (Fig. 6d), LLL8 establishes contacts with the N-terminal half of the peptide mainly through the CDR1α and CDR3α loops, whereas the CDR3β loop mostly contacts the C-terminal half.

### Interaction of TCR LLL8 with HLA-A2

Of the total number of contacts (84) that TCR LLL8 makes with HLA-A2, excluding the LLL peptide, CDR1α, CDR2α, and CDR3α contribute 20%, 6%, and 33%, respectively, compared with 0%, 31%, and 5% for CDR1β, CDR2β, and CDR3β, respectively (Table 1) (Fig. 6g). Hence, Vα dominates the interactions of LLL8 with MHC (54 of 84 contacts: 64%), with the somatically generated CDR3α loop contributing more than any other CDR to MHC recognition (28 contacts). TCR LLL8 makes many more interactions with the HLA-A2 α1 helix than the α2 helix (Fig. 6e, f), mainly through CDR3α and CDR2β. These include a dense network of six hydrogen bonds linking Gln96α, Tyr48β, and Gln50β to Arg65H and Gln72H of helix α1 (Supplementary Table 4). In addition, Arg28α forms two hydrogen bonds with Glu166H at the C-terminus of helix α2 that further anchor LLL8 to HLA-A2. In agreement with this analysis, computational alanine scanning mutagenesis with Rosetta[57] of MHC residues in the interface with TCR identified Arg65H, Gln72H, and Glu166H as the three most energetically important HLA-A2 residues for engaging LLL8 (Supplementary Table 5).

Based on the TCR3d database of experimentally determined TCR–pMHC structures[58], there are >40 structures containing TCRs that possess the TRAV12-2 germline gene and that bind HLA-A2, collectively representing at least 10 unique human TCRs. Several of these, including the TCR A6−Tax−HLA-A2 complex (PDB code 1AO7)[59] and TCR DMF5−MART-1−HLA-A2 complex (3QDG)[60], have α chain interactions with MHC, as well as with peptide backbone, that are highly

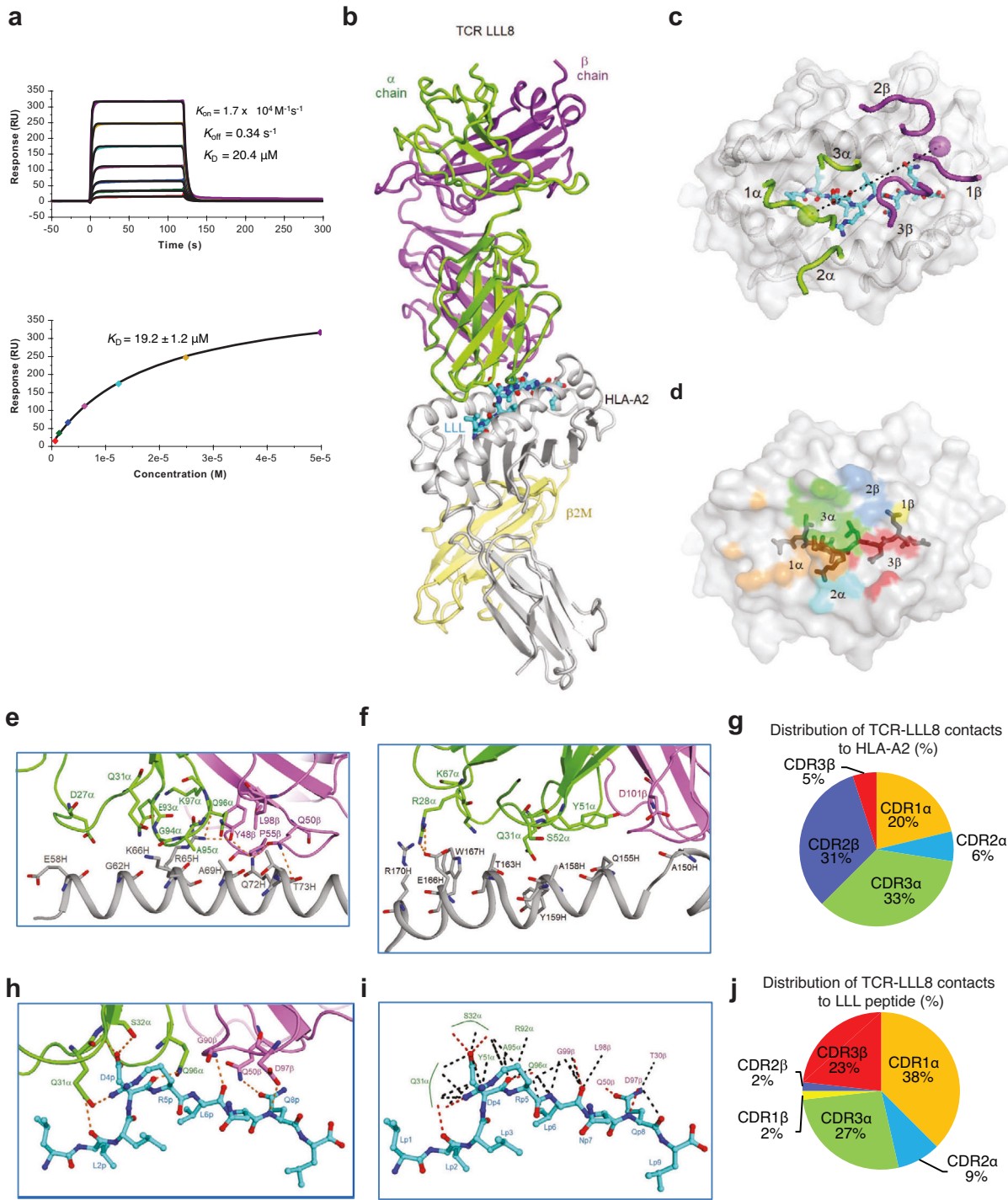

**Fig. 6 | Affinity and structure of the TCR LLL8−LLL−HLA-A2 complex. a** SPR analysis of TCR LLL8 binding to LLL−HLA-A2. TCR LLL8 at concentrations of 0.78, 1.56, 3.12, 6.25, 12.5, 25.0, and 50.0 µM injected over immobilized LLL−HLA-A2 (1200 RU). Kinetic fitting gave a $K_D$ of 20.4 µM. Fitting curve for equilibrium binding that resulted in a $K_D$ of 19.2 ± 1.2 µM. **b** Structure of the TCR LLL8−LLL−HLA-A2 complex. TCR α chain, green; TCR β chain, violet; HLA-A2 heavy chain, gray; β2microglobulin (β2m), yellow; and LLL peptide, cyan. **c** Positions of CDR loops of TCR LLL8 on LLL−HLA-A2 (top view). CDRs of LLL8 are shown as numbered green (CDR1α, CDR2α, and CDR3α) or violet (CDR1β, CDR2β, and CDR3β) loops. HLA-A2 is depicted as a gray surface. The LLL peptide is cyan. The green and violet spheres mark the positions of the conserved intrachain disulfide of the Vα and Vβ domains, respectively. The black dashed line indicates the crossing angle of TCR to pMHC. **d** Footprint of TCR LLL8 on LLL−HLA-A2. The areas contacted by individual CDR

loops are color-coded: CDR1α, orange; CDR2α, cyan; CDR3α, green; CDR1β, yellow; CDR2β, blue; CDR3β, red. **e** Interactions between TCR LLL8 and the HLA-A2 α1 helix. The side chains of contacting residues are drawn in stick representation with carbon atoms in green (TCR α chain), violet (TCR β chain) or gray (HLA-A2). Hydrogen bonds are red dotted lines. **f** Interactions between TCR LLL8 and the HLA-A2 α2 helix. **g** Pie chart showing percentage distribution of TCR LLL8 contacts to HLA-A2 according to CDR. **h** Interactions between TCR LLL8 and the LLL peptide. The side chains of contacting residues are drawn in stick representation with carbon atoms in green (TCR α chain), violet (TCR β chain) or cyan (LLL). **i** Schematic representation of TCR LLL8−LLL interactions. Hydrogen bonds are red dotted lines and van der Waals contacts are black dotted lines. For clarity, not all van der Waals contacts are shown. **j** Pie chart showing percentage distribution of TCR LLL8 contacts to LLL peptide according to CDR.

**Table 1 | LLL8 TCR atomic contacts with the LLL peptide and HLA-A2**

| | CDR1α | CDR2α | HV4α | CDR3α | CDR1β | CDR2β | HV4β | CDR3β | Total |
|---|---|---|---|---|---|---|---|---|---|
| *# Contacts* | | | | | | | | | |
| Peptide | 21 | 5 | 0 | 15 | 1 | 1 | 0 | 13 | 56 |
| HLA-A2 | 17 | 5 | 4 | 28 | 0 | 26 | 0 | 4 | 84 |
| Total | 38 | 10 | 4 | 43 | 1 | 27 | 0 | 17 | 140 |
| *% Contacts* | | | | | | | | | |
| Peptide | 38 | 9 | 0 | 27 | 2 | 2 | 0 | 23 | 100 |
| HLA-A2 | 20 | 6 | 5 | 33 | 0 | 31 | 0 | 5 | 100 |
| Total | 27 | 7 | 3 | 31 | 1 | 19 | 0 | 12 | 100 |

similar to those of TCR LLL8 (Supplementary Fig. 6b–e). These conserved interactions, which occur between germline-encoded CDR1 and CDR2 loops and pMHC, appear to support the hypothesis that the canonical diagonal docking orientation of TCR on MHC, which is maintained in the LLL8–LLL–HLA-A2 complex, is the result of coevolution of TCR and MHC molecules[61,62]. However, there are several HLA-A2-binding TCRs that possess the TRAV12-2 germline gene but whose α chains engage pMHC through different sets interactions, as seen in TCR–pMHC complex structures RD1–MART-1–HLA-A2 (5E9D)[63], 868–SL9–HLA-A2 (5NME)[64], NYE-S1–NY–ESO–1–HLA-A2 (6RPB)[65], and YLQ7-YLQ-HLA-A2 (7N1F)[25] which contains a TCR bound to a SARS-CoV-2 spike epitope (see Discussion). Thus, convergent or preferred germline interaction motifs, as observed for LLL8 and other TRAV12-2 TCRs, are not always observed and are dependent on the TCR context (CDR3, TRBV gene) and/or epitope target.

## Vα dominates LLL peptide recognition

A remarkable feature of LLL-specific TCRs isolated from COVID-19 convalescent patients is the almost exclusive use of members of TRAV12 gene family (TRAV12-2 in the case of LLL8). Coincidentally, the large majority (~85%) of HLA-A*02:01-restricted TCRs specific for the YLQ spike epitope, which is unrelated in sequence to LLL, also use TRAV12-2 or TRAV12-1 gene segments[15,26]. The TRAV12-2 chain of LLL-specific TCRs can pair with multiple Vβs, including TRBV9, 2, 7–2, 6–6, 18, and 14. TRBV gene usage appears to be widely distributed, with TRBV9 the most frequent (8.6%) out of 510 unique LLL-specific TCRs. The structure of the LLL8–LLL–HLA-A2 complex revealed the basis for this combinatorial diversity. Of the 56 total contacts that LLL8 establishes with the LLL peptide, the bulk (41; 73%) are mediated by Vα (Table 1). This Vα dominance allows pairing with multiple Vβs, which, like TRBV7-2 of LLL8, are expected to make comparatively few interactions with the peptide, as well as MHC (see above). CDR1α, CDR2α, and CDR3α account for 38%, 9%, and 27% of contacts with LLL, respectively, compared to 2%, 2%, and 23% for CDR1β, CDR2β, and CDR3β, respectively (Fig. 6j). Of note, the germline-encoded CDR1α loop contributes more than any other CDR to peptide recognition, with Gln31α and Ser32α forming a cluster of four hydrogen bonds with LLL: Gln31α Nε2–O P2 Leu, Gln31α Oε1–Nη2 P5 Arg, Ser32α N–Oδ2 P4 Asp, and Ser32α Oγ–Oδ2 P4 Asp (Fig. 6h, i) (Supplementary Table 6). It appears that the TRAV12-2 sequence is uniquely suited to providing this configuration of hydrogen bonds for specific binding with the ionic P4 Asp-P5 Arg core of the LLL peptide.

Both TRAV12-1 and TRAV12-2 encode CDR1α residues Gln31α and Ser32α, whereas TRAV12-3 encodes CDR1α residues Gln31α and Tyr32α. Computational mutagenesis of Ser32α to Tyr in the LLL8–LLL–HLA-A2 complex using Rosetta shows a highly unfavorable ΔΔ$G$ (17 kcal/mol), indicating that the TRAV12-3-encoded CDR1α Tyr residue would be incompatible or much less compatible with the LLL8 mode of LLL–HLA-A2 engagement. The TRAV12-1 CDR2α loop has a different length than TRAV12-2: 8 residues (TRAV12-1) vs. 9 residues (TRAV12-2) based on TCR3d CDR loop definitions[58]. This difference in

length leads to a preferred backbone conformation observed in most structurally characterized TRAV12-1 TCRs (e.g., PDB codes 6VRM, 7N6E, 7PBE, 7EA6) that is distinct from TRAV12-2 TCRs, including LLL8, suggesting that TRAV12-1 is incompatible with LLL8-like recognition of LLL-HLA-A2 (which includes two CDR2α residues as binding hotspots; Supplementary Table 6). Thus, residue and length features of the CDR1α and CDR2α loops of TRAV12-3 and TRAV12-1, respectively, may be responsible for the observed lack of those germline genes in LLL–HLA-A2-specific TCRs.

TCR LLL8 engages six residues of the LLL peptide, burying 353 Å$^2$ of peptide surface (Fig. 6h; Supplementary Table 7). However, most interactions involve central residues P4 Asp and P5 Arg (36 of 54 van der Waals contacts) (Fig. 6i), whose protruding side chains pack against the CDR1α and CDR3α loops. Based on SARS-CoV-2 sequences in the GISAID database[66], the LLL epitope is highly conserved, and there are only two polymorphisms with >0.1% frequency: Q229H and L230F. Analysis of the LLL8–LLL–HLA-A2 structure using Rosetta[57] predicts that the Q229H substitution at TCR-contacting position P8 will lead to maintained or improved LLL8 binding, whereas the L230F substitution at MHC anchor position P2 will prevent epitope presentation by HLA-A2. LLL8 represents a pan-sarbecovirus reactive TCR due to the conservation of the LLL epitope within that group, while in other coronaviruses (e.g., DLLNRLQAL in MERS-CoV N; four substitutions from the SARS-CoV-2 LLL sequence) it varies, and we anticipate no LLL8 cross-reactivity due to substitutions in three TCR-contacting peptide residues.

The key CDR3 residues in the LLL8–LLL–HLA-A2 complex provide insights into the observed CDR3 motifs, particularly for CDR3α. TCR LLL8 exemplifies the CDR3α motif (G/N) (G/A)(Q/N)K with its subsequence GAQK, which includes residues Ala95α and Gln96α that have key contacts with pMHC; both of those residues are identified as binding hotspots based on Rosetta (Supplementary Table 7). The specific CDR3α subsequence GAQK was observed in 27 out of 516 (5%) of LLL–HLA-A2-binding TCRs. Due in part to the apparent diversity of CDR3β sequences in LLL-specific TCRs and resultant lack of pronounced motifs, it is not clear whether the one CDR3β hotspot residue that was identified by Rosetta (Asp97β) based on the LLL8–LLL–HLA-A2 structure corresponds to a motif position and residue that is structurally conserved among LLL–HLA-A2-binding TCRs. The LLL8 CDR3β may be one example of a highly variable array of CDR3β recognition strategies in the context of restricted TCRα sequences and variable TRBV germline genes.

Superposition of the MHC α1α2 domains of unbound LLL–HLA-A2 (7KGQ)[49] onto those of LLL–HLA-A2 in complex with TCR LLL8 showed small yet relevant differences in peptide conformation, corresponding to r.m.s.d. of 0.87 Å for main-chain atoms of LLL. The largest displacement by far is for P5 Arg, whose α-carbon position shifts 2.7 Å. It appears that several residues of the LLL8 TCR α chain impinge on P5 Arg and cause it to bend from its erect posture above the peptide in unbound LLL–HLA-A2, downward and toward the HLA-2 α2 helix in the LLL8–LLL–HLA-A2 complex.

**Transcriptome alteration associated with the LLL-specific CD8+ T cell response**

In vitro stimulation with LLL-HLA-A2 separated donors into two groups: expanders who had clear expansion of LLL-recognizing CD8+ T cells and non-expanders who did not have expansion of LLL-recognizing CD8+ T cells despite the presence of detectable levels of LLL tetramer+ CD8+ T cells. To understand what regulates activation-induced proliferation, we analyzed LLL tetramer+ unstimulated CD8+ T cells from both expanders and non-expanders by scRNAseq and identified six subsets ($T_N$, $T_{SCM}$, $T_{CM}$, $T_{EM}$, $T_{EMRA}$, and activated) of CD8+ T cells (Fig. 7a, b, Supplementary Fig. 7a) based on their characteristic gene expression features. Next, we compared the transcriptome of CD8+ T cells and of each subset between expanders and non-expanders using GSEA and found that CD8+ T cells ($T_N$, $T_{SCM}$, $T_{EM}$, and $T_{EMRA}$) from expanders expressed enriched genes involved in negative regulation of chemotaxis, cytokine activity, and responses to calcium ions (Supplementary Fig. 7b), whereas CD8+ T cells ($T_N$, $T_{SCM}$, $T_{EM}$, and $T_{EMRA}$) from non-expanders had enriched genes involved in chromatin modification, histone binding, positive regulation of cytokine production, and regulation of innate immune response (Fig. 7c). To rule out the possibility that differences in TCR quality between the two groups contributed to the activation-induced CD8+ T cell expansion, we selected LLL+ CD8+ T cells with high LLL-binding TCRs based on RF scores (>0.8) and compared the transcriptomes of the same subsets between expanders and non-expanders using GSEA. We found that these enriched functional groups presented in Figs. c, d remained the same between these two groups, suggesting that transcriptome changes identified between expanders and non-expanders are not due to differences in TCR quality. Furthermore, the enriched gene functional groups had a high degree of sharing among different CD8+ T cell subsets and are closely interact with each other as revealed by the gene network/pathway analysis (Fig. 7d, Supplementary Data 3). Together, these findings suggest a common underlying mechanism that regulates activation induced CD8+ T cell proliferation and expansion.

## Discussion

The importance of CD8+ T cells in combatting SARS-CoV-2 infection is increasingly being recognized. In this study, we show that SARS-CoV-2 infection augmented CD8+ T cell immunity against epitopes derived from the conserved N protein. The improved CD8+ T cell response includes (1) a mild increase in circulating epitope-recognizing CD8+ T cells but substantially more expansion in response to stimulation in vitro, (2) long-lasting activity over one year after infection without obvious change with age, (3) restricted Vα gene usage by TCRs recognizing LLL, and (4) shared transcriptome features associated with weaker activation-induced proliferation. These findings identified LLL as a dominant nucleocapsid epitope, characterized LLL-specific TCRs in structural terms, and revealed CD8+ T cell transcriptome features associated with expanders and non-expanders. Such information will be valuable for further evaluation of CD8+ T cell response to SARS-CoV-2 and for designing better SARS-CoV-2 vaccines which contains dominant epitopes not only from the S protein but also other proteins such as N protein. Indeed, LLL is one of six SARS-CoV-2 T cell epitopes included in a COVID-19 peptide-based vaccine (CoVac-1), which induces T cell immunity is not affected by current SARS-CoV-2 variants[28].

Analysis of CD8+ T cells that recognize six epitopes from the N protein of SARS-CoV-2 in COVID-19 convalescent and unexposed HLA-A2+ individuals revealed several key features of CD8+ T cell immunity against this virus. First, there exist low frequencies of epitope-specific CD8+ T cells with both naïve and memory phenotypes in unexposed individuals, which suggests these epitope specific memory CD8+ T cells may be activated by common coronaviruses or other viruses that shared similar sequences. Second, infection with SARS-CoV-2 results in only a slight increase in the frequencies of CD8+ T cells but significantly enhances proliferation in response to stimulation. While their

enhanced proliferation is beneficial for containing initial infection, it remains to be determined if they also contribute to unintended consequences such as long COVID[67]. Third, over a one-year period, N-recognizing CD8+ T cells from convalescent donors have stable frequencies and in vitro responses to activation, which is strikingly different from IgG titers against N proteins. These findings suggest that SARS-CoV-2 infection induced better and longer-lasting CD8+ T cell immunity in convalescent than in unexposed donors. COVID-19 vaccination also induces protective CD8+ T cell immunity[68]. It is unknown whether vaccines induce comparably robust and long-lasting CD8+ T cell immunity against SARS-CoV-2 as infection[28,69]. A better understanding of CD8+ T cell immunity against SARS-CoV-2 could serve a basis for efficacious developing T cell-based vaccines[70].

Knowledge of the diversity size of antigen-specific TCR repertoires and the nature of TCR−pMHC interactions is essential to inform us about the status of T cell immunity to SARS-CoV-2. Combining tetramer staining/cell sorting and scTCRseq, we analyzed 22,727 LLL+ CD8+ T cells, and strikingly found that LLL-binding TCRs used TRAV12-2, accounting for nearly 50% of all LLL+ TCRs. Additionally, the other half of LLL+ TCRs used different TRAV genes and none of their representative TCRs showed substantial binding or signaling. Even with the tight gating on tetramer+ cell during sorting, substantial false positive TCRs remain in the scTCR dataset. This suggests that the low frequency of antigen-specific CD8+ T cells identified by positive tetramer staining contained a high portion (50% in LLL+ CD8+ T cells) of false positives. This problem was overcome by our development of a ML algorithm (RF model) that is able to identify true LLL-recognizing TCRs with good accuracy, paving the way to curate high-quality TCRs from the pool of undefined TCRs. Like all predictive ML algorithms, its accuracy relies on the quality of positive and negative training data. By using experimentally confirmed LLL-binding and non-binding TCRs as the seed in the same cluster of TCRs classified by the TCRDist program based on CDR3 amino acid sequences[71], we were able to select an adequate number of TCRs for ML training and testing. The quality of this ML algorithm will be further tested when more LLL-binding TCRs and their crystal structures become available, which will improve its accuracy even more.

Crystal structures of several TCRs from COVID-19 convalescent patients bound to two spike epitope (YLQ and RLQ) presented by HLA-A2 have been reported[23–26]. These structures include: (1) TCR YLQ7−YLQ−HLA-A2[25], (2) TCR YLQ36−YLQ−HLA-A2[26], (3) TCR NR1C−YLQ−HLA-A2[24], (4) TCR RLQ3−RLQ−HLA-A2[25], and (5) TCR RLQ7−RLQ−HLA-A2[72]. Notably, TCRs LLL8 and YLQ7 use the same Vα gene segment, TRAV12-2, which is closely related to the TRAV12-1 gene segment used by TCRs YLQ36 and NR1C. The α chains of the three YLQ-specific TCRs (YLQ7, YLQ36, and NR1C) dock similarly atop HLA-A2, as the result of partly or fully conserved interactions between germline-encoded CDR1α and CDR2α loops and the α1 and α2 helices of HLA-A2 (Supplementary Fig. 8a–d). However, the α chain of LLL8 is displaced by ~4.5 Å towards the N-terminus of the LLL peptide compared to its position in the YLQ7−YLQ−HLA-A2 and other complexes (Supplementary Fig. 8e), resulting in a different set of interactions between the CDR1α and CDR2α loops and HLA-A2. This displacement is probably dictated by the LLL peptide, which is unrelated to the YLQ peptide.

The obvious discrepancy between the high mortality of COVID-19 and robust immune response to COVID-19 vaccines in older adults remains a puzzle. Here, we did not find significant age-related changes in (1) plasma IgG titer against the N-protein, (2) frequencies and in vitro expansion of CD8+ T cells recognizing N-epitopes, and (3) longevity of CD8+ T cells recognizing N-epitopes over one year after infection. Due to the lack of severely ill COVID-19 patients in our study cohort, it is possible that we missed age-associated immune defects. In an attempt to understand the mechanisms underpinning robust and poor CD8+ T cell responses, we compared the transcriptome of LLL-recognizing

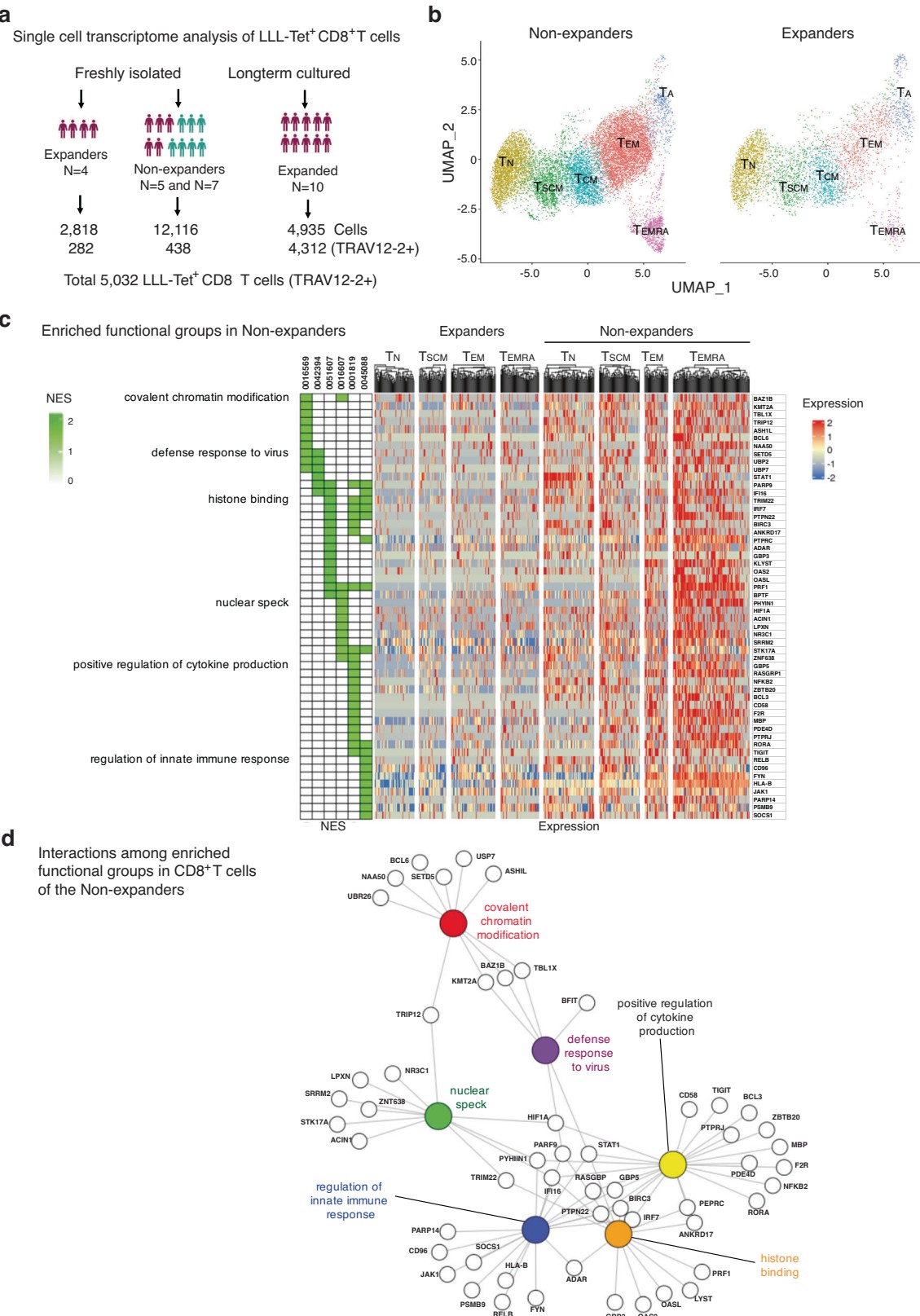

**Fig. 7 | Altered transcriptomes of CD8⁺ T cells against nucleocapsid LLL epitope in non-expanders. a** Summary of scRNAseq findings of LLL⁺ CD8⁺ T cells (total and TRAV12-2⁺) from freshly isolated cells (convalescent and uninfected donors) and in vitro stimulated cells (convalescent only). **b** UMAPs shows the subsets composition of LLL⁺ CD8⁺ T cells from expanders and non-expanders. **c** Enriched expressed genes in CD8⁺ T cells and subsets of non-expanders heatmap of selected enriched genes of CD8⁺ T cells within d0 cells of non-expanders compared to expanders. NES normalized enriched score. it ranges from 0 to 2. **d** Interactions of enriched expressed genes in CD8⁺ T cells of non-expanders. Pathway analysis of selected significantly enriched genes in CD8⁺ T cells of non-expanders in four functional GO groups.

CD8[+] T cells between expanders and non-expanders and found that CD8[+] T cells from non-expanders have enhanced expression of genes related to histone modifications (*KMT2A, PSMB9, LBH*), differentiation (*BCL6, RORA, PDCD4*), and lymphocyte-mediated immunity (*GZMB, PRF1, LYST*). These changes appear to be shared among different memory subsets. These findings suggest that advanced differentiation within the defined memory subsets is associated with poor proliferative response to peptide stimulation. In contrast, CD8[+] T cells from expanders have enhanced expression of genes related to response to calcium ions (*JUN, JUNB, DUSP1*), response to glucocorticoid (*FOS, FOSB, AIF1*), and structural molecule activity (*ACTB, ACTG1, TUBA1A*). Like what we found in non-responders, these changes are also generally shared among different CD8[+] T cell subsets. The link between these enriched genes and how they collectively facilitate better stimulation induced proliferation as well as whether such changes are associated with aging remain to be determined.

The fine specificity of TCR and the cellular competence of activation-induced proliferation and differentiation are two key elements that determine the quality of CD8[+] T cell immunity against SARS-CoV-2 and potential clinical outcomes. Empowered by single-cell technology and ML algorithms, analysis of antigen specific CD8[+] T cell response will reveal essential details of the pre-existing, post-vaccine, and post-infection status of CD8[+] T cells and will offer guidance for vaccine development and administration.

## Methods
### Human donors
Seventy-five convalescent and 138 uninfected donors were recruited under NIH IRB approved protocol (000140) and all donors provided written informed consent regarding their participation in the study. All convalescent patients had proof of positive COVID-19 PCR test and positive of anti- SARS-CoV-2 Spike protein IgG, detectable levels of anti-SARS-CoV-2 nucleocapsid IgG on the date of blood draw, and self-reported mild COVID-19 symptoms. Uninfected healthy donors who were either unvaccinated or vaccinated against SARS-CoV-2 (received the Pfizer-BioNTech or Moderna COVID-19 vaccines). All uninfected donors had undetectable levels of anti-SARS-CoV-2 nucleocapsid IgG on the date of blood draw. 78 donors did not receive the COVID-19 vaccine and had no detectable blood anti-spike and anti-nucleocapsid IgG antibodies. The other 60 donors had been vaccinated with either Pfizer-BioNTech or Moderna COVID-19 vaccines but displayed no detectable levels of anti-nucleocapsid antibodies.

### Blood processing and PBMC isolation
Blood was collected in EDTA contain tubes. Sample processing and PBMC isolation was carried out within 1–24 h of sample collection. To obtain EDTA plasma, 1.5 mL of blood was centrifuged at 438 g and the resulting plasma supernatant was collected and stored at −80 °C for antibody testing at a later date. PBMCs were isolated by diluting blood samples with Hank's Balanced Salt Solution (1X solution without calcium and magnesium), layering on Ficoll-Paque, and centrifuging at 894 g for 25 min. Cells at the interface were collected and washed twice with HBSS buffer before further processing or cryopreservation. A fraction of the isolated PBMCs were used for lymphocyte staining. The HLA-A2 genotype of the donor was determined by flow cytometry using HLA-A2-FITC antibody (BioLegend). PBMCs of HLA-A2+ donors were used for AIM assay and tetramer staining (Table 2). CD8[+] T cells were positively selected using the EasySep™ Direct Human CD8[+] T Cell Isolation Kit (STEMCELL Technologies) according to the manufacturer's instructions and used for 14-day culture with peptide stimulation.

### Detection of anti-SARS-COV-2 antibody using ELISA
All donors were tested for the presence of antibodies (IgG) against SARS-CoV-2 nucleocapsid and spike proteins. Plasma samples stored at −80 °C were prepared following the manufacturer's protocols for the LEGEND MAX™ SARS-CoV-2 Nucleocapsid Human IgG ELISA Kit and the LEGEND MAX™ SARS-CoV-2 Spike S1 Human IgG ELISA Kit. The samples were analyzed by microplate reader (SpectraMax M2, Molecular Devices) and a four-parameter logistic curve was fitted using the plate standards. All samples were titrated appropriately so that the OD value ≥ 1; a sample was considered undetectable if the OD value < 1 at a 1:50 dilution.

### Flow cytometry analysis of CD8[+] T cells
For all donors, 2 M freshly isolated PBMCs were used to analyze overall immune markers within lymphocyte populations. Cells were stained with Fixable Viability Stain 780 (BD Biosciences) at a 1:100 dilution for 5 min at room temperature, washed, and resuspended in Brilliant Stain Buffer (BD Biosciences). A surface stain cocktail mix including CD95-PE/Cy5, CD45RA-BUV805, HLA-DR-BUV737, CD8-BUV496, CD27-BUV395, CD28-BV786, CD127-BV711, CD69-BV650, CD137-BV605, CD3-V500, CD38- PerCP-eFluor 710, CD62L-FITC, and CD4-BUV661 was added (Table 2). After 30 min of incubation at 4 °C, cells were washed and fixed in 4% paraformaldehyde overnight. The next day, cells were washed using 1X Perm/Wash buffer (BD Biosciences), stained with an intracellular antibody cocktail made up of Granzyme B-PE/Cy7 and Perforin-PE/Dazzle 594, and incubated at 4 °C for 30 min. Cells were washed, resuspended in 1% paraformaldehyde, and analyzed by flow cytometry (FACSymphony, BD Biosciences). All collected flow cytometry data were analyzed by FlowJo10.5.

### Tetramer staining
All MHC class I tetramers were made by the NIH Tetramer Core Facility. Up to five tetramer-peptide reagents with contrasting fluorescence were used in a given staining cocktail. The amount of each tetramer was titrated (0.1–1 µL) to obtain the optimal concentration for usage. Freshly isolated and 14-day cultured cells were washed with PBS, stained with tetramer cocktail in PBS + 2% FBS, and incubated at 4 °C for 30 min. An antibody cocktail comprised of CD8-PerCP/Cy5.5, CD45RA-BV510, CD62L-PE/Cy7, CD95-PE/Dazzle 594 (Biolegend) was added and samples incubated for an additional 30 min at 4 °C. Samples were then washed with FACS buffer and analyzed by flow cytometry (CytoFLEX, Beckman Coulter).

### AIM assay
PBMCs (1 × 10⁶ per well) were cultured for 24 h in the presence of SARS-CoV-2 S1-specific peptides (1 mg/mL) (JPT Peptides) and six peptides of N protein (AlanScientific.com), 0.5% DMSO (equimolar amount) or 2 mg/mL phytohemagglutinin (PHA) in 96-wells U bottom plates. After stimulation, cells were collected and resuspended in 50 mL BSM. Human TruStain FcX™ Fc (2.5 µl) was added and incubated for 10 min at room temperature. Antibodies (CD8-BV510, CD38-APC, CD69-PECY7, CD137-PE, and HLA-DR-FITC, 2.5 µl each) were added (Table 2) and incubated for 30 min at 4 °C. Cells were then washed once with 2 mL FACS buffer and resuspended in 250 µL FACS buffer and collected using BD FACSCanto II.

### In Vitro stimulation and culture
Positively selected CD8[+] T cells from HLA-A02+ donors were used to determine their antigen-specific activation induced expansion in vitro as previously described[73]. A mixture of 0.2 million CD8[+] T cells, 2 million PBMCs, and 10 µg of each of the six nucleocapsid peptides was created and transferred to a 96-well round bottom plate at 100 µL medium per well. Each donor had three plates set up with the same peptide mix. Cells remained in culture at 37 °C for seven days with 40 µL of additional media being added on day 3 to replenish depleted nutrients.

On day 7, cells were harvested and counted, and 1 M cells were used for tetramer staining. Out of the remaining harvested cells, 6 M

## Table 2 | Source of reagents

| REAGENT OR RESOURCE | SOURCE | IDENTIFIER | DILUTION |
|---|---|---|---|
| CD3 UCHT1 BV570 | BioLegend | Cat#300436 | 1:50 |
| CD3 SP34-2 V500 | BD Biosciences | Cat#560770 | 1:20 |
| CD4 SK3 BUV661 | BD Biosciences | Cat#612962 | 1:20 |
| CD8a RPA-T8 BUV496 | BD Biosciences | Cat#612942 | 1:50 |
| CD8a RPA-T8 PerCP/Cy5.5 | BioLegend | Cat#301032 | 1:50 |
| CD27 L128 BUV395 | BD Biosciences | Cat#563815 | 1:50 |
| CD28 CD28.2 BV785 | BioLegend | Cat#302950 | 1:50 |
| CD38 HB7 PerCP-eFluor 710 | Invitrogen | Cat#46-0388-42 | 1:15 |
| CD45RA HI100 BUV805 | BD Biosciences | Cat#742020 | 1:50 |
| CD45RA HI100 BV510 | BioLegend | Cat#304142 | 1:50 |
| CD62L DREG-56 FITC | BioLegend | Cat#304804 | 1:20 |
| CD62L DREG-56 PE/Cy7 | BioLegend | Cat#304822 | 1:50 |
| CD69 FN50 BV650 | BioLegend | Cat#310934 | 1:15 |
| CD95 DX2 PE/Cy5 | BioLegend | Cat#305610 | 1:100 |
| CD95 DX2 PE/Dazzle 594 | BioLegend | Cat#305634 | 1:25 |
| CD127 A019D5 BV711 | BioLegend | Cat#351328 | 1:50 |
| CD137 4B4-1 BV605 | BD Biosciences | Cat#745256 | 1:20 |
| Granzyme B QA16A02 PE/Cy7 | BioLegend | Cat#372214 | 1:20 |
| HLA-A2 BB7.2 FITC | BD BBiosciences | Cat#343304 | 1:50 |
| HLA-DR G46-6 BUV737 | BD Biosciences | Cat#748339 | 1:15 |
| Perforin dG9 PE/Dazzle 594 | BioLegend | Cat#308132 | 1:100 |
| Hashtag 1 Antibody | BioLegend | Cat#394601 | 1:04 |
| Hashtag 2 Antibody | BioLegend | Cat#394603 | 1:04 |
| Hashtag 3 Antibody | BioLegend | Cat#394605 | 1:04 |
| Hashtag 4 Antibody | BioLegend | Cat#394607 | 1:04 |
| Hashtag 5 Antibody | BioLegend | Cat#394609 | 1:04 |
| Hashtag 6 Antibody | BioLegend | Cat#394611 | 1:04 |
| Hashtag 7 Antibody | BioLegend | Cat#394613 | 1:04 |
| Hashtag 8 Antibody | BioLegend | Cat#394615 | 1:04 |
| ALNTPKDHI HLA-A*02:01 Alexa Fluor 680 | NIH Tetramer Core | N/A | 1:100 |
| GMSRIGMEV HLA-A*02:01 PE | NIH Tetramer Core | N/A | 1:100 |
| ILLNKHIDA HLA-A*02:01 Alexa Fluor 680 | NIH Tetramer Core | N/A | 1:100 |
| ILLNKHIDA HLA-A*02:01 BV421 | NIH Tetramer Core | N/A | 1:100 |
| LALLLLDRL HLA-A*02:01 APC | NIH Tetramer Core | N/A | 1:500 |
| LLLDRLNQL HLA-A*02:01 APC | NIH Tetramer Core | N/A | 1:100 |
| LLLDRLNQL HLA-A*02:01 BV421 | NIH Tetramer Core | N/A | 1:100 |
| LQLPQGTTL HLA-A*02:01 PE | NIH Tetramer Core | N/A | 1:100 |
| Fixable Viability Stain 780 | BD Biosciences | Cat#565388 | 1:10 |
| LEGEND MAX™ SARS-CoV-2 | BioLegend | Cat#448107 | |
| LEGEND MAX™ SARS-CoV-2 Spike | BioLegend | Cat#447807 | |
| Ficoll-Paque PLUS | Cytiva | Cat#17144003 | |
| EasySep™ Direct Human CD8 + T Cell Isolation Kit | STEMCELL Technologies | Cat#19663 | |
| Brilliant Stain Buffer | BD Biosciences | Cat#563794 | |
| Perm/Wash Buffer | BD Biosciences | Cat#554723 | |

## Table 2 (continued) | Source of reagents

| REAGENT OR RESOURCE | SOURCE | IDENTIFIER | DILUTION |
|---|---|---|---|
| Opti-MEM™ I Reduced Serum Medium | Thermo Fisher Scientific | Cat#11058021 | |
| DMEM, Dulbecco's Modified Eagle Medium | Thermo Fisher Scientific | Cat#11965092 | |
| FuGENE® HD Transfection Reagent | Promega | Cat#E2311 | |
| Chromium Next GEM Single Cell 5' Reagent Kits v2 (Dual Index) | 10X Genomics | | |

(2 M/plate) were restimulated using a microbubble loaded HLA-A2 with our SARS-CoV-2 nucleocapsid peptides and anti-CD28. To create a microbubble for each of our six epitopes, we combined 5 μg of peptide with 100 μL of HLA-A2/anti-CD28 microbubble, rotated the mixture at 4 °C for 20 min, then left it to sit at 4 °C overnight before using. The 6 M cells were combined with 10 μL of the microbubble mixture for each of the six epitopes and spun at room temperature for 20 min before being plated onto a 96-well round bottom plate (100 μL/well). Cells remained in culture at 37 °C for seven more days with 40 μL of additional media being added on day 10 to replenish depleted nutrients.

On day 14, cells were counted and 1 M cells were taken for tetramer staining. Samples displaying antigen specific CD8+ T cell expansion were suspended in freezing media and stored in liquid nitrogen freezer. The degree of expansion was calculated by the fold change over the course of the 14-day culture (the absolute cell count of epitope-specific CD8+ T cells on day 14 divided by the absolute cell count of epitope-specific CD8+ T cells on day 0).

### scRNAseq and scTCRseq

**Single-cell library generation and sequencing analysis.** Single-cell RNA-seq (scRNA-seq) and single-cell TCR-seq (scTCR-seq) libraries were prepared following the protocol for the 10X Genomics Chromium Next GEM Single Cell 5' Reagents Kits v2 (Dual Index). Prior to generation of Gel Beads-in-emulsion (GEMs), cells were stained with LLL-tetramer (BV421 or APC) and sorted by Molflow sorter. Up to 10,000 single cells were used for each library. Some libraries were also stained with a hashtag oligo antibody to allow pooling multiple samples in one library. In brief, each cell was captured in a GEM which was then followed by reverse transcription, cleanup, and cDNA amplification. After purification of the amplified cDNA, 50 ng of the purified cDNA sample was used for GEX library and 50 ng was used to generate scTCR-seq libraries. V(D)J amplification was carried out and scTCR-seq libraries were prepared by fragmenting V(D)J segments, repairing the ends, and attaching sample indexes. Both GEX and TCR libraries was fragmented, size selected, and indexed for each library that were pooled for sequencing (Illumina Nova-Seq).

scRNA-seq FASTQ files were generated from an Illumina NovaSeq Sequencer. Read 1, Read 2, and the sample index were sequenced to 28, 91, and 8 base pairs (single index) or 10 and 10 bases (dual index), respectively. Filtered gene expression reads were mapped to human reference genome, GRCh38 2020-A via the Cellranger 7.0.0 count pipeline to obtain unique molecular identifier (UMI) counts for each individual sample. Filtered V(D)J reads were mapped to the vdj GRCh38 alts ensemble 5.0.0 reference genome via the Cellranger 7.0.0 vdj pipeline, which generated contiguous VDJ sequences per single cell. To further separate between samples, hashtag oligo libraries matching with gene expression libraries were generated by using the Cellranger 7.0.0 count pipeline. UMIs correlating with specific hashtag oligo sequences designating each sample were counted, with cells demonstrating at least 5 UMIs and a significantly higher count of UMIs for a particular sequence was labeled as the sample specified by the

particular hashtag oligo. This information was included within the meta data of gene expression libraries.

**Data integration and clustering.** Individual expression matrices were loaded in through Read10X via Seurat 4.0 and used for filtering, normalization, clustering and visualization. Cells were excluded if they expressed fewer than 500 genes, more than 10,000 genes and more than 20% mitochondrial genes. Expression was log-2-normalized via the Seurat function, NormalizeData and individual libraries saved for batch correction. All samples were merged and visualized via UMAP at 2000 features and 20 principal components to compare sample similarity. Libraries with fewer than 30 cells were merged with samples that most closely aligned within the initial clustered UMAP. Batch correction to remove potential sample to sample biases was carried out via IntegrateData, with 25 principal components and 1500 features. The batch corrected libraries were then visualized via UMAP at 25 principal components and 1500 features. Clustering was carried out via FindNeighbors and FindClusters at a resolution of 0.3 and markers defining each cluster found via FindAllMarkers to be compared and labeled by canonical single-cell markers of CD8+ T cells.

**TCR cluster and CDR3 motif analysis.** Clustering of 565 unique LLL-specific TCR sequences containing TRAV12-2 gene was performed using TCRDist clustering via CoNGA[71,74]. Unique V(D)J sequences were first collected from single-cell V(D)J libraries and sorted on frequency of appearance. V(D)J sequences were grouped on similarity in amino acid sequence Alpha V gene, CDR3A, Alpha J gene, Beta V gene, CDR3B, and Beta J gene and assigned an associated index with TCR distance based on similarity to other TCRs. The resulting distances were then Louvain clustered via the Rpackage igraph at a resolution of 1 generating 5 clusters of unique TCRs. Unique CDR3 amino acid sequences were derived LLL-TCRs based on the usage of TRAV12-2 and clusters containing experimentally confirmed LLL-HLA-A2 binding TCRs. CDR3 regions of these specific TCRs were further sorted on alpha, beta CDR3 length within the TCRDist cluster and submitted for motif analysis. CDR3 motifs were generated via the MEME Suite 5.5.

**Machine learning analysis.** To establish the machine learning model, 575 TCRs were selected from the LLL TCRs identified from scTCRseq that used TRAV12-2 (including both first and second functional alpha chain) These TCRs were then grouped using TCRDist clustering and four of TCR clusters containing with experimentally confirmed LLL-HLA-A2 binding TCRs were combined as a resulting 524 positive LLL-HLA-A2 binding TCRs. In parallel, 7719 TCRs from HLA-A2+ donors that isolated for non-LLL tetramer+ TCRs were also grouped by the TCRDist and five clusters containing with experimentally confirmed no LLL-HLA-A2 binding TCRs were combined as a resulting set of 5355 the negative TCRs. 10% of the positive and negative TCRs was withheld for testing. The remaining 90% TCRs were used to generate a dictionary of 3 amino acid long sequences (kmer) representing 5 different regions (the most left or right end Kmer as L-end and R-end, followed by left and right side kmer and center kmer) of CDR3. Ten random forest models were created using 471 positive TCRs and 471 negative TCRs from within the total negative set. Through unguided machine learning, the models generated 15 decision trees selecting kmers that accurately separated defined either the positive or negative sets. TCRs were assigned a score based on the presence of these kmers, with a score greater than 0.8 being considered a positive TCR. The validity of the models was determined by their ability to accurately separate the positive and negative set in withheld TCRs. The analysis was conducted using Python code (via scikit learn and random forest classifier).

**TCR expression and binding validation**
**Generating of pHAGE-TCRα/β plasmid.** Full-length encoding sequence of TCRα and β chains joined by the P2A "self-cleaving" site

which can terminate sequence translation at the final codon (Pro) of the 2A sequence and reinitiate translation of the following sequence. The entire sequence of TCRβ-P2A-TCRα were synthesized in pHAGE vector by Twist Bioscience.

**Lentivirus transduction of GIL-specific TCR into NJ76 cell line.** Plate HEK293 cells the day before transfection at a density of $1.5 \times 10^6$ cells per well of a 100 mm dish in 10 ml of complete growth medium (DMEM + 10% Fetal Bovine Serum). Add 17ug pHAGE-TCR, pCMV-dR8.2 and pCMV-VSV-G plasmids into 800 μl of OptiMEM together with 50 μl of FuGENE® HD reagent. Mix and incubate for 10 min at room temperature, and then add into one plate of HEK293T cells. Collected 48 h/72 h SFFV-CD8 virus (Twist Bioscience) [ref.] from the supernatant of transfected HEK293T cells. Dilute NJ76 cells into complete medium to a final concentration of $1 \times 10^6$ cells/mL with polybrene at concentration of 5 μg/ml. Add lentiviral solution to 6 mL Jurkat cells and incubate at room temperature for 20 min. Centrifuge the cells at $800 \times g$ for 30 min at 22–32 °C and remove virus containing medium. Use 6 ml media to resuspend the cell pellet, and the cells are transferred to the T25 tissue culture flask. The flask is returned to the tissue culture incubator for 2–3 days. After 3 days' culture, the NJ76 cells were stained with anti-Human TCR antibody and sorted for TCR+RFP+ double positive cells. NJ76 cells possessing GIL-specific TCRs were generated and ready for later experiments.

**Stimulation in NJ76 cell line with Nur77 GFP reporter system.** NJ76-TCR cells were cultured with the influenza GIL peptide-loaded ($10^{-8}$ M) artificial antigen presenting cells at 37 °C for 4/24 h. Anti-CD3/CD28-conjugated HLA-A2 microbubble (100–200 M/ml) were used for stimulating another aliquot of NJ76-TCR cells simultaneously in order to compare common MHC-TCR activation and GIL-specific TCR activation. The GFP expression in the TCRαβ+ NJ76-TCR cell population was quantified by Beckman CytoFLEX flow cytometry and results were analyzed with FlowJo (10.5). We later used the streptavidin-MB conjugated with biotinylated-HLA-A2/GIL and biotin-anti-CD28 for stimulation.

**Crystallographic analysis**
**Protein preparation.** Soluble TCR LLL8 for affinity measurement and structure determination was produced by in vitro folding from inclusion bodies expressed in *Escherichia coli*. Codon-optimized genes encoding the TCRα (residues 1–203) and β (1–243) chains were synthesized and cloned into the expression vector pET22b (GenScript). An interchain disulfide (CaCys157–CbCys171) was engineered to increase the folding yield of TCR αβ heterodimer. The TCR α and β chains were expressed separately as inclusion bodies in BL21(DE3) *E. coli* cells (Agilent Technologies). Bacteria were grown at 37 °C in LB medium to $OD_{600} = 0.6$–0.8 and induced with 1 mM isopropyl-b-D-thiogalactoside for 3 h. The bacteria were harvested by centrifugation and resuspended in 50 mM Tris-HCl (pH 8.0) containing 0.1 M NaCl and 2 mM EDTA. After sonication, inclusion bodies were washed three times with 50 mM Tris-HCl (pH 8.0) and 5% (v/v) Triton X-100, then dissolved in 8 M urea, 50 mM Tris-HCl (pH 8.0), 10 mM EDTA, and 10 mM DTT. For in vitro folding, the TCRα (45 mg) and TCRβ (35 mg) chains of dissolved inclusion bodies were mixed and diluted into 1-liter folding buffer containing 100 mM Tris-HCl (pH 8.0), 5 M urea, 0.4 M L-arginine-HCl, 3.7 mM cystamine, and 6.6 mM cysteamine. After dialysis at 4 °C against distilled water and 10 mM Tris-HCl (pH 8.0) for 24 and 48 h, respectively, the folding mixture was concentrated 20-fold and dialyzed overnight against 50 mM MES buffer (pH 6.0). After removal of the precipitate by centrifugation, the folding mixture was dialyzed overnight at 4 °C against 20 mM Tris-HCl (pH 8.0) and 20 mM NaCl. Disulfide-linked TCR LLL8 was purified using consecutive Superdex 200 (20 mM Tris-HCl (pH 8.0), 20 mM NaCl) and Mono Q (10 mM Tris-HCl (pH 8.0), 0–1.0 M NaCl gradient) FPLC columns (GE Healthcare).

Soluble HLA-A2 loaded with LLL peptide (LLLDRLNQL) was prepared by in vitro folding of *E. coli* inclusion bodies as described[75]. Correctly folded LLL–HLA-A2 complexes were purified using sequential Superdex 200 (20 mM Tris-HCl (pH 8.0), 20 mM NaCl) and Mono Q columns (10 mM Tris-HCl (pH 8.0), 0–1.0 M NaCl gradient). To produce biotinylated HLA-A2, a C-terminal tag (GGGLNDIFEAQKIEWHE) was attached to the HLA-A*0201 heavy chain. Biotinylation was carried out with BirA biotin ligase (Avidity).

**Crystallization and data collection.** For crystallization of the LLL8–LLL–HLA-A2 complex, TCR LLL8 was mixed with LLL–HLA-A2 in a 1:1 ratio and concentrated to 13 mg/ml. Crystals were obtained at room temperature by vapor diffusion in hanging drops. The LLL8–LLL–HLA-A2 complex crystallized in 0.1 M Tris-HCl (pH 8.5) and 13.5% (w/v) PEG 20 K. Before data collection, crystals were cryoprotected with 20% (w/v) glycerol and flash cooled. X-ray diffraction data were collected at beamline 23-ID-B of the Advanced Photon Source, Argonne National Laboratory. Diffraction data were indexed, integrated, and scaled using the program AIMLESS[76]. Data collection statistics are shown in Table S6.

**Structure determination and refinement.** The LLL8–LLL–HLA-A2 structure was determined using the molecular replacement program PHASER[76] within the CCP4i suite of crystallographic software[77] after synchrotron diffraction screening of ~100 crystals and molecular replacement searches in four of the best datasets. The successful searches used probes derived from published PDB structures 2UWE[78] and 6VRM[75]. An additional key probe, a sequence-based model of the VαVβ component of the TCR, was generated by the TCR structure prediction resource TCRmodel[79]. With four TCR–pMHC complexes (~3300 amino acids) per asymmetric unit and 3.18 Å resolution data, molecular replacement was a process of building up the solution domain-wise, first locating the MHC components, then placing the Vα and Vβ domains of the TCRs, and finally the Cα and Cβ domains. Molecular replacement outputs were evaluated for their capacity to be reproduced subject to variations in probe, dataset, and resolution shell, as well as their structural reasonableness (e.g., polypeptide continuity at domain interfaces and avoidance of steric clashes). When all MHC and TCR components had been correctly placed, refinement using REFMAC[80] lowered the *R*-free metric from 0.45 to 0.40, and difference maps showed the remaining domains, thus demonstrating that the structure was solved. From that point, maps also guided the placement of about 200 residues that differed structurally or sequence-wise from the probes or had been omitted. The last parts to be built were the LLL peptides and CDR loops in the four complexes. Final electron density was unambiguous for all the main chain, but a few side chains in the CDRs retained weak density and were confirmed by residue-specific omit-refine maps. The four final complexes in the asymmetric unit are very similar, superposing with all six pairwise root-mean-square difference (r.m.s.d.) values under 1.5 Å for α-carbon positions. The LLL8–LLL–HLA-A2 complex with the clearest maps, which also has the lowest r.m.s.d. from the other three, has been assigned chain identifiers ABDEF, is designated the biological unit, and is described in Results. Refinement statistics are summarized in Table S6. Contact residues were identified with the CONTACT program in CCP4i[77] and were defined as residues containing an atom 4.0 Å or less from a residue of the binding partner. The PyMOL program (https://pymol.org/) was used for r.m.s.d. calculations, graphical map interpretation and model building, and to prepare figures.

**Surface plasmon resonance analysis.** The interaction of TCR LLL8 with LLL–HLA-A2 was assessed by surface plasmon resonance (SPR) using a BIAcore T100 biosensor at 25 °C. Biotinylated LLL–HLA-A2 was immobilized on a streptavidin-coated BIAcore SA chip (GE Healthcare) at around 1000 resonance units (RU). The remaining streptavidin sites were blocked with 20 µM biotin solution. An additional flow cell was injected with free biotin alone to serve as a blank control. For analysis of TCR binding, solutions containing different concentrations of LLL8 were flowed sequentially (50 µl/min, 600 s for dissociation) over chips immobilized with LLL–HLA-A2 or the blank. Dissociation constants ($K_D$s) were calculated by fitting equilibrium and kinetic data to a 1:1 binding model using BIA evaluation 3.1 software.

**Computational sequence and structural analysis.** Computational mutagenesis was performed using the "interface" mode of Rosetta (v. 2.3)[81] as described previously[82], which models the mutant residue and calculates predicted energy change ($\Delta\Delta G$) of TCR–pMHC binding using an optimized energy function. For mutations to amino acids other than alanine or glycine, minimization of proximal residues was permitted ("-min_interface -int_chi" flags in Rosetta) to allow for local side chain movements to accommodate the side chain substitution. Prior to computational mutagenesis calculations, the LLL8–LLL–HLA-A2 complex structure was pre-processed using the FastRelax protocol in Rosetta 3 (weekly release 2021.38)[82], to perform constrained minimization to remove minor structural aberrations that would potentially bias subsequent Rosetta calculations. The flags used for FastRelax minimization, run with the "relax" executable in Rosetta, are noted below:

    -relax:constrain_relax_to_start_coords
    -relax:ramp_constraints false
    -ex1
    -ex2
    -use_input_sc
    -no_his_his_pairE
    -no_optH false
    -flip_HNQ

**Statistical analysis**

Group differences between convalescent and uninfected donors for total CD8+ T cells and each sub-populations were compared using separate linear regression models with each subpopulation of T-cells as the outcome. The main predictor of the model was group, with covariates of age and sex. Statistical trends with time since diagnosis for convalescent donors were analyzed via a mixed effect model accounting for multiple visits per donor with covariates of age and sex. *P* values less than 0.05 were considered significant. Two-tailed *T* tests comparing difference between convalescent and uninfected donors were normalized for age and sex. All analyses were performed using R version 4.1.0 through the stats package.

**Reporting summary**

Further information on research design is available in the Nature Portfolio Reporting Summary linked to this article.

## Data availability

Source data are provided with this paper. The scRNAseq data have been deposited in the NCBI accession code GSE227971. Atomic coordinates and structure factors for the LLL8–LLL–HLA-A2 complex have been deposited in the Protein Data Bank accession code 8DNT. Source data are provided with this paper.

## Code availability

The script files of ML model used for LLL-TCR determination are deposited at GitHub (https://github.com/Weng-lab-NIH/RF_models).

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

## Acknowledgements

This work was supported by the Intramural Research Program of the National Institutes of Health, National Institute on Aging (to N-P.Weng), by National Institutes of Health Grants GM126299 (to B.G.Pierce), GM144083 (to B.G.Pierce), and AI129893 (to R.A.Mariuzza), R44CA265468 (to G.Shi) and by National Natural Science Foundation of China Grant 32100985 (to D.Wu). Structure results in this report are based on work performed at the GM/CA beamline at the Advanced Photon Source of Argonne National Laboratory, which is funded by the National Cancer Institute (ACB-12002) and the National Institute of General Medical Sciences (AGM-12006, P30GM138396). This work utilized computational resources of the NIH HPC Biowulf cluster (http://hpc.nih.gov) and the University of Maryland Institute for Bioscience and Biotechnology Research High Performance Computing Cluster. We acknowledge the NIAID Tetramer Core Facility for providing HLA-A2 tetramers. Identification of commercial materials and equipment does not imply recommendation or endorsement by the National Institute of Standards and Technology.

## Author contributions

C.C., J.L. and J.L. carried out most wetlab experiments. J.C. conducted most scRNAseq and scTCRseq analysis and statistical analysis. H.H. and J.C. assisted computational analysis, A.Z., Q.Y., B.B. and E.W. assisted cell isolation and/or flow analysis, J.M., D.M., J.R. and L.Z., were responsible of recruiting donors and obtain blood sample, T.W., C.D., and C.N. assisted flow cytometry analysis and cell sort, C.W.C., L.F.,

J.M.E. assisted clinical data, protocol and advice, G.S. and Y-T.L. assisted with microbubble for in vitro stimulation, T.A. and M.S.H.K. designed peptide pools of SARS-CoV-2 proteins, A.Y. and O.M. helped statistical analysis, J.S.F., W.W.W., C-K.C., R-F.S. and S.D. helped sequencing, R.W. provided technical assistance of making tetramers. D.T.G. and D.W. performed the crystallography and structural analyses, and R.Y. performed structural analysis and modeling. B.G.P. and R.A.M. conceived and supervised the structure project. J.M.E. L.F. and N-P.W. conceived and supervised the project. C.C., J.C., D.T.G., R.A.M. and N-P.W. wrote the manuscript. All authors prepared the paper.

## Funding

## Competing interests

Y-T.L. is the founder of Diagnologix LLC and G.S. works for Diagnologix LLC. M.S.H.K. is the founder of Elixirgen Therapeutics, Inc and T.A. works for Elixirgen Therapeutics, Inc. The rest of authors declare no conflict of interests.

## Additional information

Cecily Choy [1,13], Joseph Chen[1,13], Jiangyuan Li [1], D. Travis Gallagher[2], Jian Lu[1], Daichao Wu [3], Ainslee Zou [1], Humza Hemani [1], Beverly A. Baptiste[1], Emily Wichmann[1], Qian Yang[1], Jeffrey Ciffelo[1], Rui Yin [3], Julia McKelvy[4], Denise Melvin[4], Tonya Wallace[1], Christopher Dunn [1], Cuong Nguyen[1], Chee W. Chia[4], Jinshui Fan[5], Jeannie Ruffolo[6], Linda Zukley[6], Guixin Shi[7], Tomokazu Amano[8], Yang An[9], Osorio Meirelles[10], Wells W. Wu[11], Chao-Kai Chou[11], Rong-Fong Shen[11], Richard A. Willis [12], Minoru S. H. Ko [8], Yu-Tsueng Liu[7], Supriyo De [5], Brian G. Pierce [3], Luigi Ferrucci [6], Josephine Egan [4], Roy Mariuzza [3] & Nan-Ping Weng [1,13] ✉

[1]Laboratory of Molecular Biology and Immunology, National Institute on Aging, NIH, Baltimore, MD, USA. [2]National Institute of Standards and Technology (NIST), Gaithersburg, MD, USA. [3]W.M. Keck Laboratory for Structural Biology, University of Maryland Institute for Bioscience and Biotechnology Research, Rockville, MD, USA. [4]Laboratory of Clinical Investigation, National Institute on Aging, NIH, Baltimore, MD, USA. [5]Computational Biology and Genomics Core, Laboratory of Genetics and Genomics, National Institute on Aging, NIH, Baltimore, MD, USA. [6]Translational Gerontology Branch, National Institute on Aging, NIH, Baltimore, MD, USA. [7]Diagnologix LLC, San Diego, CA, USA. [8]Elixirgen Therapeutics, Inc, Baltimore, MD, USA. [9]Laboratory of Behavioral Neuroscience, National Institute on Aging, NIH, Baltimore, MD, USA. [10]Laboratory of Epidemiology & Population Sciences, National Institute on Aging, NIH, Baltimore, MD, USA. [11]Facility for Biotechnology Resources, CBER, Food and Drug Administration, Silver Spring, MD, USA. [12]NIH Tetramer Core Facility at Emory University, Atlanta, GA, USA. [13]These authors contributed equally: Cecily Choy, Joseph Chen, Nan-Ping Weng. ✉e-mail: wengn@mail.nih.gov

