## [Peer Review File · Nature Communications]

REVIEWER COMMENTS

Reviewer #1 (expert in expert in TCR structure and X-ray crystallography):

This is a good study on cellular immunity against SARS-CoV-2 infection. The authors took multiple approaches to investigate a dominant epitope (LLDRLNQL, referred to LLL) from the nucleocapsid (N) protein of the virus identified in both convalescent and uninfected donors that is presented by the HLA-A2 to be recognized by CTL cells. They found through scTCRseq analysis that the LLL-specific CTL cells use highly restricted Va genes that can pair with relatively diverse Vb genes in LLL+ CTL cells. They found that although the increase of circulating epitope-recognizing CTL cells is mild, but substantially more expansion in response to stimulation in vitro is observed, which is regulated by gene expression from transcriptome analysis.

The authors also combined functional analyses with structural scrutiny of LLL-specific TCR LLL8 in complex with LLL peptide presented by HLA-A2. They employed machine learning algorithm to accurately predict LLL-HLA-A2 binding TCRs. They determine crystal structure of LLL-HLA-A2-TCR. The structure clearly shows the Va demination. There are multiple hydrogen bonds from TCR to both peptide (particularly with the hotspot D4-R5) and HLA molecule from Va, in particular via CDR1a and CDR3a, which beautifully explains the restricted Va gene usage. By comparison, Vb portion involves much less specific interaction with pMHC, mostly to peptide Q8 residue. The works are comprehensive and solid. It provides much needed investigation on cellular immunity against SARS-CoV-2 that warrants its publication in Nature Communications.

I have three minor comments for the authors to consider. 1) the authors studied epitope from nucleocapsid protein. In literature there are also epitopes (like YLQ) from spike protein presented by the same HLA-A2. It will be interesting for the authors to carry out some structural comparison. 2) In line 307, the authors mentioned coevolution of TCR and MHC molecules and cited some papers. Since this is a still controversial issue, the authors should rephrase it. 3) the authors essentially used "contact" to describe the inter-molecular interactions. I would prefer to use more hydrogen bonding as specific interaction.

Reviewer #2 (expert in T cell responses to viruses, including SARS-CoV2):

This is an interesting report that undertakes a detailed analysis of nucleocapsid-specific CD8+ T cells.

The work is very well presented.

Points

- a major finding is that there are detectable T cells in patients who are SARS-CoV-2 uninfected. Indeed, these are in the memory pool. I presume that it is assumed that these are cross-reactive with the other endemic coronaviruses? No mention is made of this and it should be discussed. Indeed I think the authors should assess and compare relative homology in these peptides between the viruses

- the observation that the CD8+ T cell repertoire is altered after SARS-CoV-2 is interesting. Is the implication that there are substantial proportions of SARS-CoV-2 specific CD8+ T cells? Or could this be a bystander effect - and potentially be related to issues such as Long Covid ? More could be made of this in the Discussion

Reviewer #3 (expert in TCR sequencing):

In their manuscript "SARS-CoV-2 infection establishes an enhanced, stable, and age-independent CD8+ T cell response against a dominant nucleocapsid epitope using highly restricted TCRs", Choy and colleagues analyze the response of convalescent and uninfected donors to 6 epitopes from the nucleocapsid of the SARS-CoV-2. They show an enhanced response to the dominant epitope LLL in convalescent donors as well as a sustainable response over the time. They also demonstrate that

TCR recognizing the LLL-HLA-A2 complex is strongly bias toward the TRAV12-2 V segment, a finding explained by the importance of the alpha chain in the binding of the LLL-HLA-A2 complex. Finally, they demonstrated that the difference in the magnitude of the response to stimulation with LLL peptide might be due to variation in the differentiation status of the CD8 T cells. The manuscript is well writing and easy to read. However, I still have some points and comments that have to be addressed to validate their finding.

Major comments :

1. Figure 1e-f : The authors claim that a higher frequency of CD8 specific T cells was observed in convalescent patient only for the ALN and LLL peptides. However, supplementary table 2 shows that for the peptides LQL, LAL and GSM, tetramer staining were detected in 83% of the convalescent donors, but only in 25% of the uninfected donors. This difference could not be ignore to determine the impact of SARS-CoV-2 infection on the presence of specific CD8 T cells, and donors with no positive signal should not be remove from the analysis. A value of 0 should be used in case of absence of staining.

Interestingly, the difference in the percentage of positive donors between convalescent and uninfected patient was not observed for the ALN, LLL and ILL peptides. Could the authors comment on the difference observed between this two groups of peptides. Would cross reactivity with other coronavirus be an explanation of this discrepancy?

2. Figure 2 and S2: Unless I misunderstood the figure, I am not convinced that the conclusion drawn in figure 2 and S2 are supported by the data. It seems that, like for figure 1, the analyses do not take in account time point with tetramer negative staining. Looking at the data for the frequency of ALN+ CD8 T cells, it is clearly visible that most of the patients do not have any positive signal at their third visit. The higher number of positive patients for IgG and ILL+ CD8 T cell during the third visit suggest that samples were collected and analyzed. If this is true, it cannot be concluded that epitope specific CD8 T cells remain stable over the year if the detection of those cells have been lost at the third visit for 80 to 90% of the patients. Data should be reanalyzed including all patients, using a 0 value for the time point where no signal was detected.

3. Figure 4 : I found the observation that the response to SARS-CoV-2 is not related to the age very interesting. However, and in line with the previous comments, are the time point with no positive staining included for the average calculation? If not, the figure should be reanalyzed including these values as the absence of detection suggest a diminution of the response.

4. Figure 5 : The data presented in figure 5 are demonstrating a strong bias of the repertoire. However, it would be interesting to know how many unique abTCR were detected for each population (expanders, non expander and long-term cultured) (figure 5a). Furthermore, showing separately the V distribution for the freshly isolated cells and long-term culture could be helpful. Indeed, according to figure 5a, the TRAV12-2 frequency reach 86% in the long-term culture against 7 to 9% in the freshly isolated cells, further highlighting the strong enrichment of TRAV12-2 following stimulation.

The authors should also consider analyzing the data at the TCR level. What are the most expanded clonotypes in the long-term culture and what are their frequencies. Are these TCRs also present in the freshly isolated cells and at which frequencies? If those expanded TCRs are found in the non-expanders population, it could strength the transcriptomic data showing that proliferation is not related to the TCR sequence but to cell states.

5. From what I understood, clustering and motif analyses are based only on the number of TCR without considering their frequencies. Taking in account the frequencies may completely change the importance of a cluster or a motif in the recognition of an epitope. Therefore, having both the number and the frequency values for the figure 5d and 5e would be very useful to evaluate the impact and the importance of each cluster/motif in the recognition of the epitope.

The total percentage of the table in figure 5e, does not correspond to the sum of the column. Could the authors clarify how these percentages were calculated.

6. In line with the previous comment, the GAQK modified was observed in 5% of the TCRs and seems to provide an optimal binding to the peptide. What is the frequency of the TCR have the GAQK motif, are their the most expanded TCRs ?

7. As mentioned in point 4, if authors could show that TCRs that expand in the long-term culture are also present in the freshly isolated cells of the non-expanders, it will be a nice orthogonal validation of the figure 7c-d.

Minor comments :

1. Figure 1f : in the text, the authors say "TCM CD8+ T cells specific for LLL and N315-324(GMSRIGMEV, referred to as GMS) were significantly increased in convalescent patient". However, figure shows significant difference for ALN and GMS. The text and the figure should be harmonized.

2. Figure S3a : value below 0 for a percentage of cells is a bit strange. All negative values should be set to 0.

3. Figure 3d : I don't understand why the authors say that they found a positive correlation for TCM and not for TN, TSCM, TEM and TERMA as the S- and p-value are the same for all subpopulations. Could the authors provide more information or a R2 value to help the reader ?

4. Figure 4b and S4a : Regarding the 2 log difference from the values between figure 4a, S4a and figure 1e, I guess that the legend of the y-axis for the figure 4b and S4a is not in percent as indicated.

RESPONSE TO REVIEWERS' COMMENTS

Reviewer #1 (expert in TCR structure and X-ray crystallography):

This is a good study on cellular immunity against SARS-CoV-2 infection. The authors took multiple approaches to investigate a dominant epitope (LLLDRLNQL, referred to LLL) from the nucleocapsid (N) protein of the virus identified in both convalescent and uninfected donors that is presented by the HLA-A2 to be recognized by CTL cells. They found through scTCRseq analysis that the LLL-specific CTL cells use highly restricted Va genes that can pair with relatively diverse Vb genes in LLL+ CTL cells. They found that although the increase of circulating epitope-recognizing CTL cells is mild, but substantially more expansion in response to stimulation in vitro is observed, which is regulated by gene expression from transcriptome analysis.

Response: *We are grateful to the reviewer for his/her appreciation of our study and for calling our attention to the following important points:*

1. The authors studied epitope from nucleocapsid protein. In literature there are also epitopes (like YLQ) from spike protein presented by the same HLA-A2. It will be interesting for the authors to carry out some structural comparison.

Response: *The reviewer is correct that there are crystal structures of TCRs bound to two spike epitopes (RLQ and YLQ) presented by HLA-A2. These structures are: 1) TCR YLQ7–YLQ–HLA-A2 (7N1F), 2) TCR YLQ36–YLQ–HLA-A2 (7PBE), 3) TCR NR1C–YLQ–HLA-A2 (7N6E), 4) TCR RLQ3–RLQ–HLA-A2 (PDB code 7N1E), and 5) TCR RLQ7–RLQ–HLA-A2 (8GOM). We have added these other structures to the Discussion (p. 20, 2nd paragraph):*

“Crystal structures of several TCRs from COVID-19 convalescent patients bound to two spike epitope (YLQ and RLQ) presented by HLA-A2 have been reported^{23,24 25,26}. These structures include: 1) TCR YLQ7–YLQ–HLA-A2²⁵, 2) TCR YLQ36–YLQ–HLA-A2²⁶, 3) TCR NR1C–YLQ–HLA-A2²⁴, 4) TCR RLQ3–RLQ–HLA-A2²⁵, and 5) TCR RLQ7–RLQ–HLA-A2. Notably, TCRs LLL8 and YLQ7 use the same V α gene segment, TRAV12-2, which is closely related to the TRAV12-1 gene segment used by TCRs YLQ36 and NR1C. The α chains of the three YLQ-specific TCRs (YLQ7, YLQ36, and NR1C) dock similarly atop HLA-A2, as the result of partly or fully conserved interactions between germline-encoded CDR1 α and CDR2 α loops and the α 1 and α 2 helices of HLA-A2 (Supplementary Figure 8a-d). However, the α chain of LLL8 is displaced by ~4.5 Å towards the N-terminus of the LLL peptide compared to its position in the YLQ7–YLQ–HLA-A2 and other complexes (Supplementary Figure 8e), resulting in a different set of interactions between the CDR1 α and CDR2 α loops and HLA-A2. This displacement is probably dictated by the LLL peptide, which is unrelated to the YLQ peptide.”

2. In line 307, the authors mentioned coevolution of TCR and MHC molecules and cited some papers. Since this is a still controversial issue, the authors should rephrase it.

Response: *We agree that the idea of coevolution of TCR and MHC remains controversial and thank the reviewer for making this point. Accordingly, we have rewritten the paragraph on coevolution to present a more balanced view (p. 14, 1st paragraph):*

“Based on the TCR3d database of experimentally determined TCR–pMHC structures⁵⁷, there are >40 structures containing TCRs that possess the TRAV12-2 germline gene and that bind HLA-A2, collectively representing at least 10 unique human TCRs. Several of these, including the TCR A6–Tax–HLA-

A2 complex (PDB code 1A07)⁵⁸ and TCR DMF5–MART-1–HLA-A2 complex (3QDG)⁵⁹, have α chain interactions with MHC, as well as with peptide backbone, that are highly similar to those of TCR LLL8 (Supplementary Figure 6b-e). These conserved interactions, which occur between germline-encoded CDR1 α and CDR2 α loops and pMHC, appear to support the hypothesis that the canonical diagonal docking orientation of TCR on MHC, which is maintained in the LLL8–LLL–HLA-A2 complex, is the result of coevolution of TCR and MHC molecule^{60 61}. However, there are several HLA-A2-binding TCRs that possess the TRAV12-2 germline gene but whose α chains engage pMHC through different sets of interactions, as seen in TCR–pMHC complex structures RD1–MART-1–HLA-A2 (5E9D)⁶², 868–SL9–HLA-A2 (5NME), NYE-S1–NY–ESO–1–HLA-A2 (6RPB)⁶³, and YLQ7–YLQ–HLA-A2 (7N1F)²⁵ which contains a TCR bound to a SARS-CoV-2 spike epitope (see Discussion). Thus, convergent or preferred germline interaction motifs, as observed for LLL8 and other TRAV12-2 TCRs, are not always observed and are dependent on the TCR context (CDR3, TRBV gene) and/or epitope target.”

3. The authors essentially used “contact” to describe the inter-molecular interactions. I would prefer to use more hydrogen bonding as specific interaction.

Response: We did not intend to downplay the known role of hydrogen bonds as determinants of specificity. We have added a new sentence (underlined) to our description of TCR–peptide interactions to emphasize this point (p. 15, 1st paragraph):

“Of note, the germline-encoded CDR1 α loop contributes more than any other CDR to peptide recognition, with Gln31 α and Ser32 α forming a cluster of four hydrogen bonds with LLL: Gln31 α N ϵ 2–O P2 Leu, Gln31 α O ϵ 1–N η 2 P5 Arg, Ser32 α N–O δ 2 P4 Asp, and Ser32 α O γ –O δ 2 P4 Asp (Figure 6h-i) (Supplementary Table 11). It appears that the TRAV12-2 sequence is uniquely suited to providing this configuration of hydrogen bonds for specific binding with the ionic P4 Asp-P5 Arg core of the LLL peptide.”

Reviewer #2 (expert in T cell responses to viruses, including SARS-CoV2):

This is an interesting report that undertakes a detailed analysis of nucleocapsid-specific CD8+ T cells. The work is very well presented.

Response: We are grateful to the reviewer for his/her appreciation of our study and for calling our attention to the following important points:

1. a major finding is that there are detectable T cells in patients who are SARS-CoV-2 uninfected. Indeed, these are in the memory pool. I presume that it is assumed that these are cross-reactive with the other endemic coronaviruses? No mention is made of this and it should be discussed. Indeed I think the authors should assess and compare relative homology in these peptides between the viruses

Response: The pre-existing six N-epitope specific CD8 T cells contained both naïve (~20%) and memory (~80%) phenotype cells in uninfected donors. The memory CD8 T cells are likely cross-reactive with other viruses. Although sequence analysis of LLL epitope showed little overlap with other common

coronaviruses (new Supplementary Table 3), this does not exclude crossing with other type of viruses. We have discussed this in revised manuscript (p6, 2nd paragraph and p19 2nd paragraph).

2. the observation that the CD8+ T cell repertoire is altered after SARS-CoV-2 is interesting. Is the implication that there are substantial proportions of SARS-CoV-2 specific CD8+ T cells? Or could this be a bystander effect - and potentially be related to issues such as Long Covid? More could be made of this in the Discussion

Response: *Thanks for raising this point. In vitro stimulation resulting in expansion of selected TCRs could be reflect the range quality of TCRs that can bind to the specific tetramer and/or the quality of the cells to undergo a robust proliferative response. Whether some of these selected TCRs contribute to Long COVID is interesting and will require additional study. We have discussed this in revised manuscript (p19, 2nd paragraph).*

Reviewer #3 (expert in TCR sequencing):

In their manuscript “SARS-CoV-2 infection establishes an enhanced, stable, and age-independent CD8+ T cell response against a dominant nucleocapsid epitope using highly restricted TCRs”, Choy and colleagues analyze the response of convalescent and uninfected donors to 6 epitopes from the nucleocapsid of the SARS-CoV-2. They show an enhanced response to the dominant epitope LLL in convalescent donors as well as a sustainable response over the time. They also demonstrate that TCR recognizing the LLL-HLA-A2 complex is strongly bias toward the TRAV12-2 V segment, a finding explained by the importance of the alpha chain in the binding of the LLL-HLA-A2 complex. Finally, they demonstrated that the difference in the magnitude of the response to stimulation with LLL peptide might be due to variation in the differentiation status of the CD8 T cells.

The manuscript is well writing and easy to read. However, I still have some points and comments that have to be addressed to validate their finding.

Response: *We are grateful to the reviewer for his/her appreciation of our study and for calling our attention to the following important points:*

Major comments:

1a. Figure 1e-f: The authors claim that a higher frequency of CD8 specific T cells was observed in convalescent patient only for the ALN and LLL peptides. However, supplementary table 2 shows that for the peptides LQL, LAL and GSM, tetramer staining were detected in 83% of the convalescent donors, but only in 25% of the uninfected donors. This difference could not be ignore to determine the impact of SARS-CoV-2 infection on the presence of specific CD8 T cells, and donors with no positive signal should not be remove from the analysis. A value of 0 should be used in case of absence of staining.

Response: *We agree that 0.00% value is important to be included and we did this in our original manuscript. Those blank cells in Supplementary Table 2 are no data collected (now adding N/A), not a lack of positive signal (0.00%). During the course of this study, we analyzed a total of 24 epitopes and*

only a subset of the total epitope-tetramers were used each time due to tetramers panel changing or a lack of comparable fluorescent in a mix. During this review, we have conducted additional staining of some samples with missing data as well as new visit 3 samples collected after our submission. The newly collected data are presented in the revised Figure 1e-f.

1b. Interestingly, the difference in the percentage of positive donors between convalescent and uninfected patient was not observed for the ALN, LLL and ILL peptides. Could the authors comment on the difference observed between this two groups of peptides. Would cross reactivity with other coronavirus be an explanation of this discrepancy?

Response: *Szeto et al had compared and found low similarity of N protein sequences between SARS-CoV-2 and four common coronavirus (ref 47). Here, we further compared these six epitope sequences with four common coronaviruses and also did not find substantial similarity (see new Supplemental Table 3). In the revised Figure 1e with newly collected data, we found that only CD8⁺ T cells to LLL remained significantly different between two groups. Cross reactivity of existing TCRs is a likely to explain the similarity in CD8 T cell frequency between convalescent and uninfected donors for the majority of N-epitopes. We have discussed this in the revised manuscript (p6, 2nd paragraph and p19 2nd paragraph).*

2a. Figure 2 and S2: Unless I misunderstood the figure, I am not convinced that the conclusion drawn in figure 2 and S2 are supported by the data. It seems that, like for figure 1, the analyses do not take in account time point with tetramer negative staining. Looking at the data for the frequency of ALN+ CD8 T cells, it is clearly visible that most of the patients do not have any positive signal at their third visit.

Response: *Please see the response to Comment 1, blank cells are not 0.00% value but rather missing data. You are right about fewer third visit data of some epitopes. We have now completed analysis of these missing third visits using frozen PBMCs and also included some third visits samples which we collected post-submission (missing visit(s) of 22 convalescent donors with total 29 samples were included in the revised Figure 1 and 2 and corresponding Supplementary figure 1 and 2). In agreement with our original submission, we did not observe a reduction in frequency of these N-epitopes binding CD8⁺ T cells overtime.*

2b. The higher number of positive patients for IgG and ILL+ CD8 T cell during the third visit suggest that samples were collected and analyzed. If this is true, it cannot be concluded that epitope specific CD8 T cells remain stable over the year if the detection of those cells have been lost at the third visit for 80 to 90% of the patients. Data should be reanalyzed including all patients, using a 0 value for the time point where no signal was detected.

Response: *As described in the above comment, with the addition of new third visited data, the revised Figure 2 and Supplementary Figure 2 show that epitope-specific CD8 T cells remain stable overtime for all six epitope specific CD8 T cells.*

3. Figure 4: I found the observation that the response to SARS-CoV-2 is not related to the age very interesting. However, and in line with the previous comments, are the time point with no positive staining included for the average calculation? If not, the figure should be reanalyzed including these

values as the absence of detection suggest a diminution of the response.

Response: *Donors with no positive staining (0.00%) are included in the analysis. As with Comments 1 and 2, donors who were stained with a given tetramer yet had no positive signal have a 0.00% value in Supplementary Table 2 while blank cells indicate that the donor was not stained for that particular tetramer. We thank the reviewer for the comment.*

4a. Figure 5: The data presented in figure 5 are demonstrating a strong bias of the repertoire. However, it would be interesting to know how many unique abTCR were detected for each population (expanders, non expander and long-term cultured) (figure 5a). Furthermore, showing separately the V distribution for the freshly isolated cells and long-term culture could be helpful. Indeed, according to figure 5a, the TRAV12-2 frequency reach 86% in the long-term culture against 7 to 9% in the freshly isolated cells, further highlighting the strong enrichment of TRAV12-2 following stimulation.

Response: *We have now included number of unique ab TCR and their corresponding cells of three groups (Expander, Non-expander and Expanded) as well as the number of TRAV12-2+ cells. in the revised Fig5a.*

4b. The authors should also consider analyzing the data at the TCR level. What are the most expanded clonotypes in the long-term culture and what are their frequencies. Are these TCRs also present in the freshly isolated cells and at which frequencies? If those expanded TCRs are found in the non-expanders population, it could strength the transcriptomic data showing that proliferation is not related to the TCR sequence but to cell states.

Response: *We do not have data of most expanded TCR clone since we do not have the frequency at the beginning of culture. The most abundant TCRs are individually different which is difficult to compare among different donors due to no prior knowledge of their frequency. As CD8 T cells isolated from 25-35 ml blood of each donor, the coverage of the true antigen-specific TCR repertoire is limited. This reflects low overlapping TCRs in donors cross three groups. There are nly 2 TCRs across three groups corresponding to 18 cells, 5 TCRs shared between Expander and Expanded corresponding to 2587 cells, and 13 between Expander and Non-expander corresponding to 213 cells. For this reason, we used the shared TCR clusters, rather than identical TCRs, for comparison in transcriptomic data.*

5a. From what I understood, clustering and motif analyses are based only on the number of TCR without considering their frequencies. Taking in account the frequencies may completely change the importance of a cluster or a motif in the recognition of an epitope. Therefore, having both the number and the frequency values for the figure 5d and 5e would be very useful to evaluate the impact and the importance of each cluster/motif in the recognition of the epitope.

Response: *This is an important point if the TCRs from the cells of the same conditions. As our data are mixed with freshly isolated and in vitro expanded TCRs, such analysis will be highly biased with expanded TCRs, which will not provide an intended result.*

5b. The total percentage of the table in figure 5e, does not correspond to the sum of the column. Could the authors clarify how these percentages were calculated.

Response: *We have corrected this error*

6. In line with the previous comment, the GAQK motif was observed in 5% of the TCRs and seems to provide an optimal binding to the peptide. What is the frequency of the TCRs that have the GAQK motif, are their the most expanded TCRs?

Response: *The GAQK sequence containing TCRs were not expanded with same 5.0% of LLL CD8 T cells (683/13037).*

7. As mentioned in point 4, if authors could show that TCRs that expand in the long-term culture are also present in the freshly isolated cells of the non-expanders, it will be a nice orthogonal validation of the figure 7c-d.

Response: *This is an excellent point. Unfortunately, the limited overlap of TCRs did not allow us to conduct this suggested analysis.*

Minor comments:

1. Figure 1f: in the text, the authors say "TCM CD8+ T cells specific for LLL and N315-324(GMSRIGMEV, referred to as GMS) were significantly increased in convalescent patient". However, figure shows significant difference for ALN and GMS. The text and the figure should be harmonized.

Response: *With new data included, TCM CD8+ T cells specific for LLL was increased but not significant as well as other epitopes (see revised figure 1f).*

2. Figure S3a: value below 0 for a percentage of cells is a bit strange. All negative values should be set to 0.

Response: *We agree and revised Fig S3a.*

3. Figure 3d: I don't understand why the authors say that they found a positive correlation for TCM and not for TN, TSCM, TEM and TERMA as the S- and p-value are the same for all subpopulations. Could the authors provide more information or a R2 value to help the reader?

Response: *We have re-analyzed included new data. In revised Figure 3d, Tem was significant and other subsets were not significant in revised SFig. 3c.*

4. Figure 4b and S4a: Regarding the 2 log difference from the values between figure 4a, S4a and figure 1e, I guess that the legend of the y-axis for the figure 4b and S4a is not in percent as indicated.

Response: *We acknowledge that the way that the y-axis is presented in Figure 4b and Supplementary Figure 4a is difficult to interpret, and as a result, we have revised these figures so that the axes are log-*

transformed. Now all figures in question (Figure 4b, Supplementary 4a, and Figure 1e-f) represent the percentage of epitope-specific CD8 T cells on a log-transformed axis.

Sincerely yours,

Nan-ping Weng, M.D., Ph.D.
Laboratory of Molecular Biology and Immunology
National Institute on Aging, National Institutes of Health
Tel: 410 558 8341
Cell: 301 529 7384
Fax: 301 480 4838
Email: wengn@mail.nih.gov

REVIEWERS' COMMENTS

Reviewer #1 (expert in TCR structure and X-ray crystallography):

In general the authors have answered my concerns. The only comment I still have is that although the authors have compared their TCR-pMHC structure with other published similar structures, I would be more curious to see what the slight difference in TCR docking on two peptides might bear any interesting biology? I know this is often difficult to explore. But in any case if the authors can find these structural differences with immunological meaning, their work might become much more impactful.

Having said that, I must say, the current revision already warrants its publication in Nature Communications.

Reviewer #3 (expert in TCR sequencing):

The authors responded to all my comments and the additional points added for the third visit make the results of figure 2 more convincing.

However, there is still two minor points to be address :

For the figure 1, the authors mentioned that the missing points are not the lack of signal but the fact that data where not collected. Therefore, the sentence in line 134-136 "We analyzed CD8+ T cells recognizing six previously reported epitopes (presented by HLA-A2) of the highly conserved N protein of SARS-CoV in 35 convalescent and 80 uninfected controls with the HLA-A2 serotype." Should be rephrased, as the epitopes were not tested against all donors as suggested.

There is a misunderstanding with my concern of figure 4b and S4a y-axis. The problem was not the log scale but the percentage. I guess the axis should be either 0.001 or 0.1% but not 0.001%. Otherwise, the authors should explained why in figure 1e they have tetramer staining at 0.1% and in figure 4b at 0.001%.

RESPONSE TO REVIEWERS' COMMENTS

Reviewer #1 (expert in TCR structure and X-ray crystallography):

In general the authors have answered my concerns. The only comment I still have is that although the authors have compared their TCR-pMHC structure with other published similar structures, I would be more curious to see what the slight difference in TCR docking on two peptides might bear any interesting biology? I know this is often difficult to explore. But in any case if the authors can find these structural differences with immunological meaning, their work might become much more impactful.

Having said that, I must say, the current revision already warrants its publication in Nature Communications.

Response: We are grateful to the reviewer for his/her appreciation of our study. Unfortunately, we cannot correlate small differences in TCR docking on peptides with differences in biological function. We wish we could say more.

Reviewer #3 (expert in TCR sequencing):

The authors responded to all my comments and the additional points added for the third visit make the results of figure 2 more convincing. However, there is still two minor points to be address :

1. For the figure 1, the authors mentioned that the missing points are not the lack of signal but the fact that data where not collected. Therefore, the sentence in line 134-136 "We analyzed CD8+ T cells recognizing six previously reported epitopes (presented by HLA-A2) of the highly conserved N protein of SARS-CoV in 35 convalescent and 80 uninfected controls with the HLA-A2 serotype." Should be rephrased, as the epitopes were not tested against all donors as suggested.

Response: We have made the suggested changes "We analyzed CD8+ T cells recognizing six previously reported epitopes (presented by HLA-A2) of the highly conserved N protein of SARS-CoV in convalescent (n=34-35) and uninfected controls (n=21-73) with the HLA-A2 serotype". Thanks.

2. There is a misunderstanding with my concern of figure 4b and S4a y-axis. The problem was not the log scale but the percentage. I guess the axis should be either 0.001 or 0.1% but not 0.001%. Otherwise, the authors should explained why in figure 1e they have tetramer staining at 0.1% and in figure 4b at 0.001%.

Response: We thank the reviewer for pointing out our error. We have revised Fig. 4b and S4a to reflect the percentage.

Sincerely yours,

Nan-ping Weng, M.D., Ph.D.
Laboratory of Molecular Biology and Immunology
National Institute on Aging, National Institutes of Health
Tel: 410 558 8341
Cell: 301 529 7384
Fax: 301 480 4838
Email: wengn@mail.nih.gov